# Robo2 regulates synaptic oxytocin content by affecting actin dynamics

**Savani Anbalagan[1†], Janna Blechman[1†], Michael Gliksberg[1], Ludmila Gordon[1], Ron Rotkopf[2,3], Tali Dadosh[4], Eyal Shimoni[4], Gil Levkowitz[1]\***

[1]Department of Molecular Cell Biology, Weizmann Institute of Science, Rehovot, Israel; [2]Bioinformatics Unit, LSCF, Weizmann Institute of Science, Rehovot, Israel; [3]Electron Microscopy Unit, Weizmann Institute of Science, Rehovot, Israel; [4]Department of Chemical Research Support, Weizmann Institute of Science, Rehovot, Israel

**Abstract** The regulation of neuropeptide level at the site of release is essential for proper neurophysiological functions. We focused on a prominent neuropeptide, oxytocin (OXT) in the zebrafish as an in vivo model to visualize and quantify OXT content at the resolution of a single synapse. We found that OXT-loaded synapses were enriched with polymerized actin. Perturbation of actin filaments by either cytochalasin-D or conditional Cofilin expression resulted in decreased synaptic OXT levels. Genetic loss of *robo2* or *slit3* displayed decreased synaptic OXT content and *robo2* mutants displayed reduced mobility of the actin probe Lifeact-EGFP in OXT synapses. Using a novel transgenic reporter allowing real-time monitoring of OXT-loaded vesicles, we show that *robo2* mutants display slower rate of vesicles accumulation. OXT-specific expression of dominant-negative Cdc42, which is a key regulator of actin dynamics and a downstream effector of Robo2, led to a dose-dependent increase in OXT content in WT, and a dampened effect in *robo2* mutants. Our results link Slit3-Robo2-Cdc42, which controls local actin dynamics, with the maintenance of synaptic neuropeptide levels.

**\*For correspondence:**
gil.levkowitz@weizmann.ac.il

[†]These authors contributed equally to this work

**Competing interests:** The authors declare that no competing interests exist.

## Introduction

The regulation of neurotransmitter level at the site of release is essential for proper neuronal function and requires constant replenishment, capture and removal of excess or aged components in synapses. To address the fundamental topic of synaptic neuropeptide homeostasis we focused on the well-studied neuropeptide, oxytocin (OXT). Oxytocin is a classical evolutionarily conserved neuropeptide, which is involved in the maintenance of various homeostatic functions and whose major axonal release site is the posterior pituitary, also known as the neurohypophysis (*Pearson and Placzek, 2013*; *Wircer et al., 2016*). Thus, hypothalamic neurosecretory cells produce the neuropeptides vasopressin and oxytocin that are packed into large dense core vesicles (LDCV) and are transported along the axons that terminate in the neurohypophysis. Upon physiological demand, the neuropeptide is released into the blood stream to influence the function of target cells throughout the body (*Burbach et al., 2001*; *Knobloch and Grinevich, 2014*). In contrast to small neurotransmitters synaptic release, which mainly occurs in highly specialized membrane structures called active zones, neuropeptides, such as OXT, are released from LDCVs from any part of the neuron, including dendrites and *en passant* axonal synapses (*Chini et al., 2017*; *Leng et al., 2008*; *Leng and Ludwig, 2008*; *Morris and Pow, 1988*). Accordingly, axonal termini of hypothalamic magnocellular OXT neurons converge into the neurohypophysis, where they form numerous *en passant* synapses in a form of highly dense axonal varicosities, also known as axonal swellings or Herring bodies (*Tweedle et al., 1989*). These structures have been identified as *bona fide* synapses that store OXT-containing LDCV

and release them upon physiological demand (*Miyata et al., 2001*; *Wittkowski and Brinkmann, 1974*).

The mechanisms that regulate the synaptic OXT vesicles content are unknown. F-actin, one of the major cytoskeleton elements in synapses play a key role in synapse formation (*Chia et al., 2014*; *Ganguly et al., 2015*). Several recent studies reported that F-actin regulates multiple aspects of vesicular homeostasis such as presynaptic vesicular capture, clustering, docking, release, recycling and inter-synaptic exchange (*Chia et al., 2014*; *Ganguly et al., 2015*; *Guillet et al., 2016*; *Marra et al., 2012*; *Miki et al., 2016*; *Soykan et al., 2017*; *Stavoe and Colón-Ramos, 2012*; *Vincent et al., 2015*). Actin is also required for recruitment of multiple synaptic proteins and receptors that are essential for synaptic function (*Sankaranarayanan et al., 2003*). In rat neurohypophyseal synapses, EM studies have shown that actin filaments are arranged both in the synaptic cytoplasm associated with the vesicles and along the plasma membrane (*Alonso et al., 1981*). Furthermore, perturbation of isolated neurohypophyseal tissue using actin disrupting agents leads to release of OXT, suggesting that cortical actin is required to prevent release of synaptic OXT (*Tobin and Ludwig, 2007*).

Here we used a combination of transgenic OXT-specific zebrafish reporters allowing monitoring and quantification of synaptic OXT levels. We investigated the role of actin in synaptic OXT content. We show that Slit3-Robo2-Cdc42 signaling, which was previously associated with modulation of actin polymerization in the growth cones of guided axons, regulates synaptic actin dynamics and OXT neuropeptide content in neurohypophyseal termini.

## Results

### Quantitative analysis of synaptic OXT neuropeptide levels in vivo

The optically transparent zebrafish larva has a few dozens of OXT neurons, which enables analysing the function of each neuron down to the single-synapse resolution in the context of a living vertebrate animal (*Blechman et al., 2011*; *Wircer et al., 2017*; *Gutnick et al., 2011*). Because zebrafish neurohypophyseal synapses were never characterized, we firstly performed transmission electron microscopy (TEM) to visualize those synapses in zebrafish larva. To localize the neurohypophysis, we used a transgenic reporter, Tg(*oxt*:EGFP), in which neurohypophyseal OXT synaptic varicosities are filled with cytoplasmic EGFP (*Figure 1A*). We observed that neurohypohyseal synapses were enriched with multiple large dense core vesicles (LDCV) (*Figure 1B & C*). We also observed an occurrence of LDVCs content exocytosis, typical of previously reported synaptic neurosecretion in the neurohypophysis (*Boudier, 1974*; *Buma and Nieuwenhuys, 1987*; *Douglas et al., 1970*; *Douglas, 1973*; *Hayashi et al., 1994*; *Santolaya et al., 1972*). Neurohypophyseal synapses were located next to the basement membrane of the endothelial cells (*Figure 1B,C*). In particular, we observed exocytotic pits, in which the membranes of LDCVs were fused with the plasma membrane, next to the perivascular space (*Figure 1B'*, arrow). This was in contrast to axons which contained null or low number of LDCVs which were relatively smaller (*Figure 1B and C* yellow dashed lines). Moreover, we detected electron-dense plasma membrane invaginations (*Figure 1C'* red arrowhead) and recycled vesicles in the cytoplasm (*Figure 1C'*, white arrowhead) indicating a membrane retrieval process by endocytosis. Coupling of synaptic exocytosis and endocytosis is typical for neuronal synapses and considered as a general mechanism for conservation of the cell surface upon neurosecretion (*Damer and Creutz, 1994*; *De Camilli, 1995*; *Douglas, 1973*; *Kononenko and Haucke, 2015*; *Rizzoli, 2014*; *Wu et al., 2014*).

To further verify the synaptic identity of the OXT-positive neurohypophyseal axonal swellings, we utilized the transgenic synaptic vesicle reporter, Tg(UAS:Sypb-EGFP) (*Zada et al., 2014*), in combination with a transgenic OXT-specific driver line *Tg(oxt:Gal4)* to specifically mark OXT synapses (*Figure 1D*). This conditional double transgenic line, *Tg(oxt:Gal4;UAS:Sypb-EGFP)* was subsequently subjected to immuno-staining with a specific antibody directed to the mature OXT nonapeptide (cleaved, cyclised and amidated) which is enriched in neurohypophyseal termini (*Gutnick et al., 2011*). We observed that almost all the neurohypophyseal anti-OXT immunoreactive puncta co-localized with Synaptophysin-EGFP (Mean weighted colocalization coefficient = 0.87, n = 5 larvae), indicating that mature OXT neuropeptides are located exclusively within neurohypophyseal synaptic axonal swellings (*Figure 1E*).

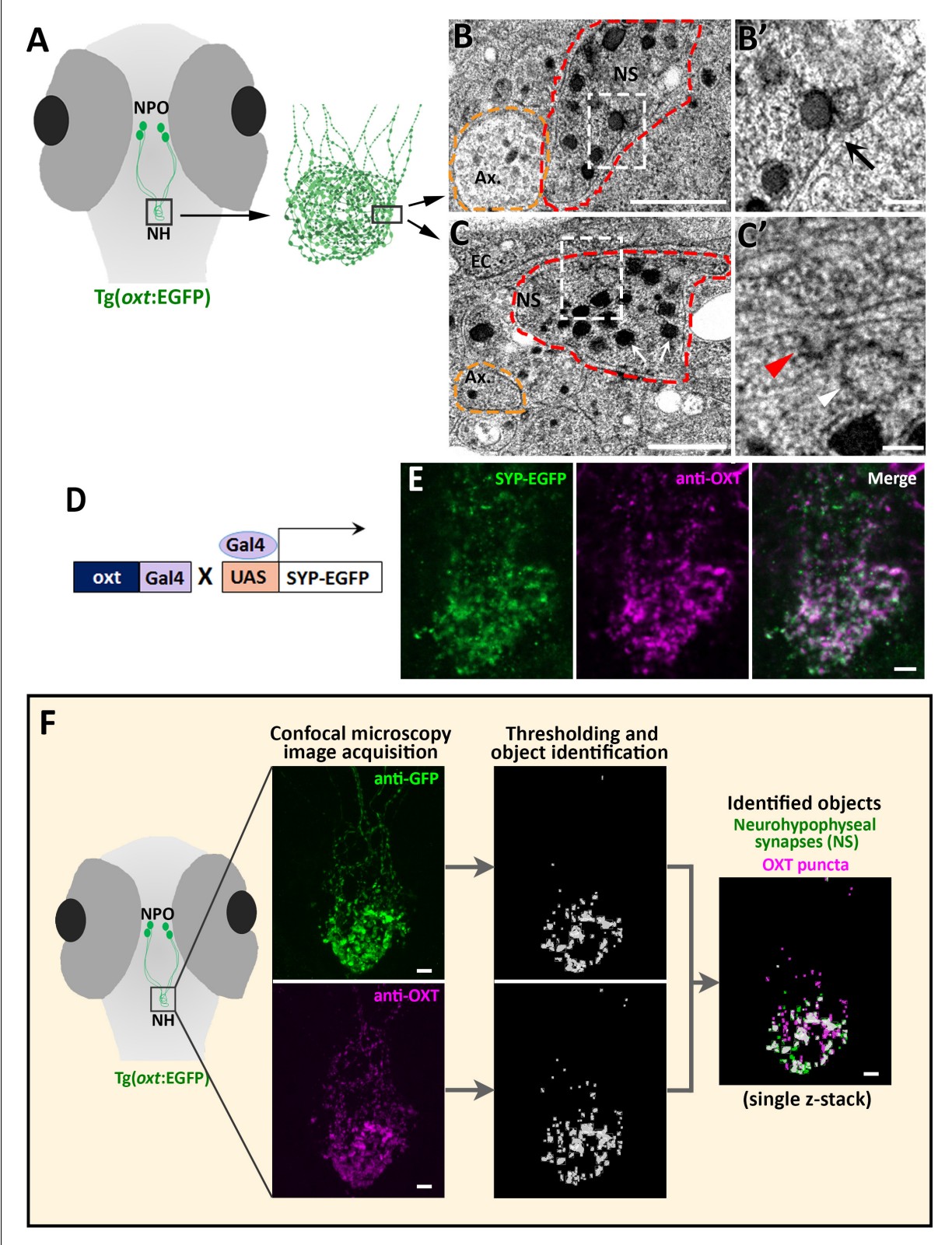

**Figure 1.** Identification and quantification of synaptic OXT content in neurohypophyseal axonal termini. (A–C) Scheme describing the neurohypophyseal vasculature and synapses in 5 days post-fertilization (dpf) transgenic reporter Tg(*oxt*:EGFP). The dense neurohypophyseal synapses (NS) were used as an anatomical landmark to localize the larval neurohypophysis prior to tissue preparation for Transmission Electron Microscopy (TEM) imaging (A). TEM image showing cross-section of axons (yellow dashed line) and neurohypophyseal synapses (red dashed line) (B,C). Large dense core

*Figure 1 continued on next page*

*Figure 1 continued*

vesicles with electron-dense neuropeptides are visible in NS (white arrows). Membranal fusion resembling exocytotic pit (black arrow), membranal invagination resembling endocytic pit (red arrow head) and empty vesicle resembling recycled vesicle (white arrow head) are visible in NS (B',C'). Ax: axon, BM: basement membrane, EC: endothelial cell and NS: neurohypophyseal synapse. Scale: 500 nM (B,C) and 100 nm (B', C'). (D,E) Oxytocin neuron-specific labeling of synapses using EGFP-fused synaptic vesicle marker synaptophysin-B (D). Whole-mount confocal microscope imaging of hypophysis of 5 days post-fertilization (dpf) transgenic Tg(*oxt*:Gal4;UAS:SYP-EGFP) zebrafish following immunostaining with anti-EGFP and a specific antibody against endogenous OXT protein (E). Mean weighted colocalization coefficients for OXT fluorescence with respect to GFP fluorescence = 0.87; n = 5 larvae (See Materials and methods for detailed description). Scale: 5 µM. (F) Scheme describing the pipeline for detection of neurohypophyseal synapses. Whole-mount confocal microscope imaging of hypophysis of 5 days post-fertilization (dpf) transgenic reporter Tg(*oxt*: EGFP) zebrafish following immunostaining with anti-EGFP and a specific antibody against endogenous OXT protein. Analysis of GFP-positive neurohypophyseal synapses (NS) and OXT puncta were performed by using the 'object identifier' function in Volocity software on individual channels. Scale: 5 µM.

The online version of this article includes the following video and source data for figure 1:

**Source data 1.** Experimental data for *Figure 1*: colocalization analysis of synaptic OXT and EGFP fluorescence.

**Figure 1—video 1.** Confocal Z-stack images and three-dimensional render of 5 days post-fertilization (dpf) transgenic reporter zebrafish Tg(*oxt*:EGFP) following immunostaining with anti-EGFP and a specific antibody against endogenous OXT protein.

https://elifesciences.org/articles/45650#fig1video1

We next visualized and quantified OXT neuropeptide content at the resolution of a single synapse by combining anti-OXT antibody staining with transgenic *Tg(oxt:EGFP)* reporter, mentioned above (*Figure 1F*). In this manner, the structure of the synapse itself, labeled by EGFP, could be differentiated from it´s content of oxytocinergic LDCVs, labeled by the anti-OXT antibody. We used image thresholding settings that allowed detection of individual EGFP-labeled synapses and their neuropeptide content, which appeared in the form of immune-reactive OXT puncta that colocalized with these EGFP-labeled synapses (*Figure 1F* and *Figure 1—video 1*).

To validate our detection method, we subjected the fish to hypertonic osmotic challenge (25% sea salt) (*Figure 2A*), which is known to induce a robust release of oxytocin and vasopressin and, consequently reduced neuropeptide content in the pituitary of both mammals and fish (*Balment et al., 1980*; *Huang et al., 1996*; *Leng and Russell, 2019*; *Pierson et al., 1995*). We detected several hundreds of synapses and OXT-stained puncta in the neurohypophysis of naïve 8 days post-fertilization (dpf) zebrafish larvae. While the number of these EGFP-positive synapses remained unaltered following acute hypertonic challenge, both the synaptic and OXT puncta volumes were decreased (*Figure 2B–E*). Recovery of larvae from hypertonic to isotonic condition (for 1 hr) led to partial-reversal of the observed phenotypes (*Figure 2C and E*). These results are in agreement with the reported hypertonicity-induced cell shrinking mechanisms essential for osmosensation in these neurons (*Prager-Khoutorsky et al., 2014*).

To further validate our method, we utilized a conditional triple transgenic line *Tg(oxt:Gal4; UAS: NTR-mCherry; UAS:BoTxLCB-GFP)* for OXT neurons-specific expression of botulinum toxin, which cleaves the vesicle-associated membrane protein (VAMP) thereby inhibiting LDCVs synaptic release. In this transgenic line, synaptic vesicle release from OXT neurons was blocked due to intracellular expression of botulinum toxin light chain B (BoTxLCB) fused to EGFP (*Sternberg et al., 2016*) and OXT synapses were visualized via mCherry (*Figure 2F*). OXT-specific expression of BoTxLCB led to increased volume of OXT puncta, but without affecting the number and volume of the synapses, indicative of accumulation of OXT content (*Figure 2G–J*). The increased synaptic OXT level we observed in BoTxLCB transgenic larvae under naïve conditions, was maintained following hypertonic challenge. These results suggest that both basal OXT release and hypertonic osmotic challenge-induced OXT release are blocked by expression of BoTxLCB. We conclude that the above methodology can faithfully quantify synaptic OXT neuropeptide content in vivo.

## Disruption of F-actin affects synaptic OXT neuropeptide content

As mentioned above, recent studies reported that dynamic changes in synaptic actin regulates multiple aspects of vesicular homeostasis. To study the spatial relationship between actin and synaptic OXT neuropeptide we performed super-resolution microscopy of a double transgenic line, *Tg(oxt: Gal4; UAS:Lifeact-EGFP)*, in which the filamentous actin (F-actin) probe, Lifeact-EGFP, was specifically expressed in OXT neurons. *Tg(oxt:Gal4; UAS:Lifeact-EGFP)* larvae were immunostained with

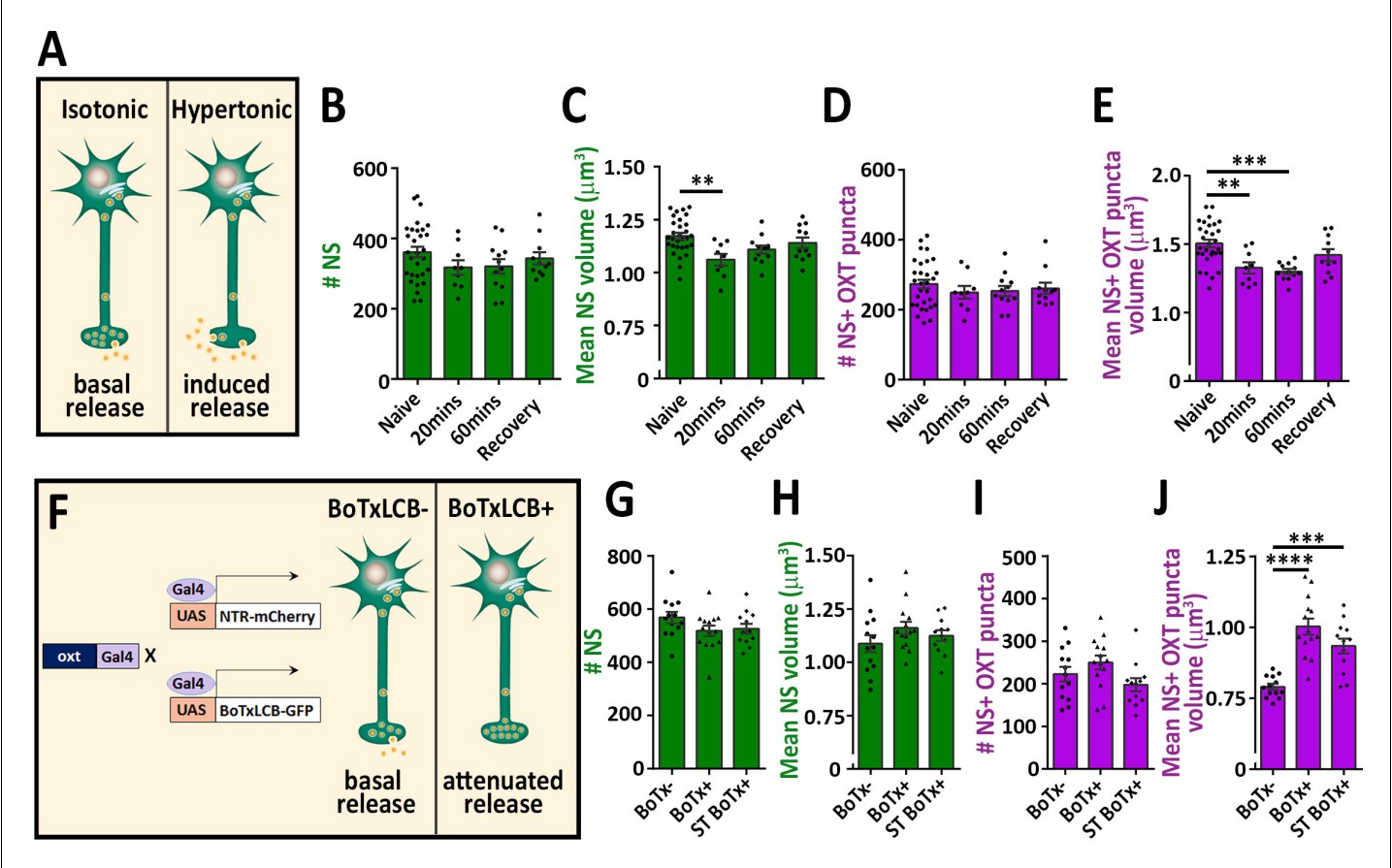

**Figure 2.** Osmotic challenge decreases synaptic OXT neuropeptide content. (**A**) Transgenic Tg(*oxt*:EGFP) larvae at eight dpf were treated with hypertonic solution (25% artificial sea salt in Danieau buffer) for 20 or 60 min. The larvae that underwent 60 min treatment were allowed to recover in isotonic Danieau buffer for another 60 min. The mean number and volume of NS (**B,C**) and OXT puncta overlapping with GFP +NS (**D,E**) were quantified (**p<0.01 and ***p<0.001; one-way ANOVA). (**F–J**) Cell-specific blockage of synaptic release using botulinum toxin light chain B increases hypophyseal OXT levels (**F**). The mean number and volume of mCherry +NS (**G,H**) and oxytocin puncta overlapping with mcherry +NS (**I,J**) were quantified in 8-dpf old larvae of Tg(*oxt*:Gal4; UAS:NTR-mCherry) labeled as BoTx- (n = 13 larvae) versus Tg(*oxt*:Gal4; UAS:BoTxLCB-GFP; UAS:NTR-mCherry) at isotonic Danieau buffer (BoTx+ (n = 14 larvae)) or upon 60 min treatment with hypertonic solution (ST BoTx+; n = 12 larvae). (***p<0.001, ****p<0.0001; one-way ANOVA). Error bars indicate SEM in (B-E and G-J).

The online version of this article includes the following source data for figure 2:

**Source data 1.** Experimental data for *Figure 2*: Neurohypophyseal synaptic parameters upon hyperosmotic challenge.

antibodies against EGFP and endogenous OXT and imaged using stochastic optical reconstruction microscopy (STORM) (*Figure 3A–D*). Imaging revealed various characteristics of actin and neuropeptides. Lifeact-EGFP signal exhibited a cage-like structure surrounding OXT in neurohypophyseal synapses (*Figure 3B–D* and *Figure 3—video 1*). These results are in agreement with previous reports that actin filaments form a cage-like structure associated with synaptic vesicles (*Hirokawa et al., 1989*; *Miyamoto, 1995*).

As Lifeact can also bind monomeric actin and it's binding to F-actin can be affected by the presence of specific actin regulators such as cofilin (*Courtemanche et al., 2016*), we undertook two alternative pharmacological and genetic approaches to link actin polymerization to synaptic OXT content. We first perturbed the actin filaments by using the actin depolymerizing agent cytochalasin D (*Cooper, 1987*). Transgenic Tg(*oxt*:EGFP) larvae were treated with cytochalasin D between 4 and 5 days post-fertilization at a concentration of 400 nM, which is a tolerated dose that allows normal development and viability of zebrafish larvae (*Trendowski et al., 2014*). Subsequent analysis revealed that the number and size of the synaptic axonal swellings was reduced upon cytochalasin D

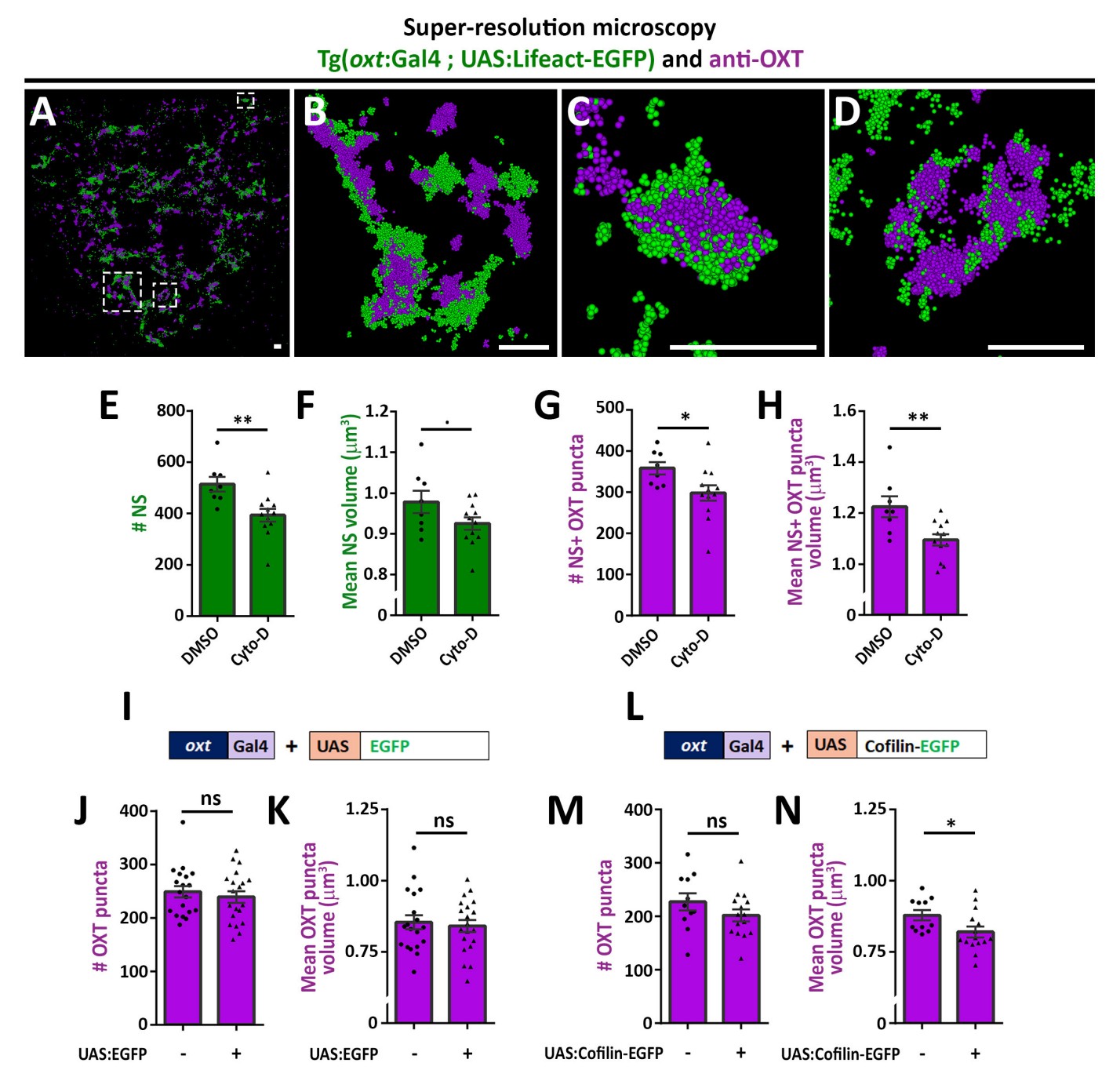

**Figure 3.** Actin polymerization regulates synaptic OXT content. (A–D) Spatial relationship between actin and neuropeptide in hypophysis revealed by super-resolution microscopy. Stochastic optical reconstruction microscopy (STORM) images of the neurohypophyseal area of 6 days post-fertilization (dpf) Tg(*oxt*:gal4; UAS:lifeact-EGFP) larvae stained using anti-OXT and anti-GFP antibodies (A). (B–D:) Magnifications of squared areas. Scale bars: 1 μm. (E–H) Assessment of the effect of cytochalasin D treatment on synaptic properties in Tg(*oxt*:EGFP) larvae. Larvae were treated with DMSO or cytochalasin D at 400 nM between 4 and 5 dpf, stained using anti-GFP and anti-OXT antibodies and quantified following imaging as in *Figure 1*. The number and mean volume of neurohypophyseal synapses (NS) (E,F) and the number and mean volume of NS-associated OXT puncta (G,H) was quantified. p<0.01, Student's *t*-test, Cohen's d = 1.54 (E); p<0.1, Student's *t*-test, Cohen's d = 0.85 (F); p<0.05, Student's *t*-test, Cohen's d = 1.17 (G); p<0.01, Student's *t*-test, Cohen's d = 1.43 (H), n = 8, 12 for DMSO and cytochalasin D treatment, respectively. (I–N) Assessment of the effect of oxytocin-neuron specific perturbation of actin. Transgenic embryos expressing the oxytocin Gal4 driver (*oxt*:Gal4) were injected with transposon-based transgenic vectors containing either control UAS:EGFP (I) or UAS:Cofilin-EGFP (L) expression cassettes. All the injected larvae were sorted for fluorescent heart selection marker and were subjected to immunostaining with anti-EGFP and anti-OXT antibody. 5-dpf larvae with and without

*Figure 3 continued on next page*

*Figure 3 continued*

expression of EGFP-labeled neurohypophyseal synapses were imaged using confocal microscopy as in *Figure 1F*. The number (**J,M**) and volume of OXT puncta (**K,N**) were quantified. (J-M: ns, non-significant; N: p<0.05; Student's t-test with Cohen's d = 0.92). Error bars indicate SEM in (E-H, J,K,M and N).

The online version of this article includes the following video and source data for figure 3:

**Source data 1.** Experimental data for *Figure 3*: Neurohypophyseal synaptic parameters upon actin perturbation.
**Figure 3—video 1.** Three-dimensional render showing STORM images of endogenous OXT and Lifeact-EGFP in neurohypophyseal synapses.
https://elifesciences.org/articles/45650#fig3video1

treatment (*Figure 3E,F*), which is in agreement with the involvement of F-actin in regulating the synaptic morphology (*Zhang and Benson, 2001*). Cytochalasin D treatment also affected both the number and volume of OXT puncta (*Figure 3G,H*).

As an alternative approach to test the role of actin dynamics, we expressed the Cofilin-EGFP fusion protein using the *Tg(oxt:Gal4)* transgenic driver, which allowed manipulation of the actin polymerization specifically in oxytocinergic neurons. Cofilin is a member of ADF family of actin-binding proteins that promote F-actin depolymerizing and regulates its turnover (*Shekhar and Carlier, 2017*; *Wioland et al., 2017*). The effects of Cofilin on OXT synapses was tested by co-injecting transposon-based DNA constructs harboring *oxt*:Gal4 together with either UAS:Cofilin-EGFP or UAS:EGFP construct as a control and thereafter monitoring OXT levels in either EGFP- or Cofilin-EGFP- positive compared to EGFP-negative OXT synapses. We observed that expression of control EGFP alone did not affect number and volume of OXT puncta however, expression of Cofilin-EGFP led to decreased OXT puncta volume (*Figure 3I–N*) with a large effect size of Cohen's d of 0.93, p-value<0.05 (*Cohen, 2013*). Together, the above results suggest that actin polymerization regulates synaptic OXT neuropeptide content.

## Robo2 regulates synaptic actin dynamics

Next, we searched for candidate signaling pathways that could regulate axonal F-actin in OXT neurons. Robo2 is localized to axonal growth cones and is known to regulate axonal guidance by modulating actin dynamics via other actin interacting regulatory proteins (*Kidd et al., 1998*; *Slováková et al., 2012*). Fluorescent in situ hybridization of *robo2* mRNA showed that it is expressed in OXT neurons (*Figure 4A*). To investigate if Robo2 regulates synaptic actin dynamics, we used the zebrafish *robo2*-deficient mutant, *astray* [*robo2*$^{272z/272z}$ (*Fricke et al., 2001*), and performed fluorescence recovering after photobleaching (FRAP) on individual OXT synapses expressing the F-actin sensor *Tg(oxt:Gal4;UAS:Lifeact-EGFP)* in either *robo2+/+* or *robo2-/-* animals (*Figure 4B, C*). We reasoned that the dynamics of Lifeact-EGFP fluorescence recovery indicates changes in synaptic polymerized actin, which is available for Lifeact-EGFP binding. Thus, it is expected that synapses wherein actin filaments are highly stable would display an increased time of Lifeact-EGFP fluorescence recovery. Indeed, in comparison to the *robo2+/+* zebrafish, the recovery of Lifeact-EGFP fluorescence was attenuated in OXT synapses of *robo2-/-* mutants (*Figure 4D*), which exhibited decreased dynamic Lifeact-EGFP fraction and increased stable fraction (*Figure 4E,F*; Cohen's d effect size 0.67, *p* value < 0.05.) This suggests that Robo2 signaling regulates synaptic actin dynamics of neurohypophyseal OXT neurons.

## Slit3-Robo2 signaling regulates synaptic OXT content

We next asked if Robo2 plays a role in synaptic accumulation of OXT by examining whether neurohypophyseal neuropeptide content is altered in the *robo2*-deficient mutant zebrafish. We found that the volume of OXT puncta was smaller in *robo2-/-* fish compared to WT controls with a large effect size (Cohen's d of 0.82, p value<0.05), while the number and size of neurohypophyseal synapses were largely unaffected, suggesting that Robo2 regulates synaptic oxytocin levels without affecting synaptic morphogenesis (*Figure 5A–D*). This phenotype was not due to OXT axonal guidance deficits, as the number of neurohypophyseal OXT axonal projections was similar between *robo2-/-* and *robo2+/+* fish larvae (*Figure 5E*).

We recently identified that *slit3*, a cognate ligand for Robo2 is highly expressed in the developing neurohypophysis (*Anbalagan et al., 2018*). We performed genetic gain/loss-of-function experiments

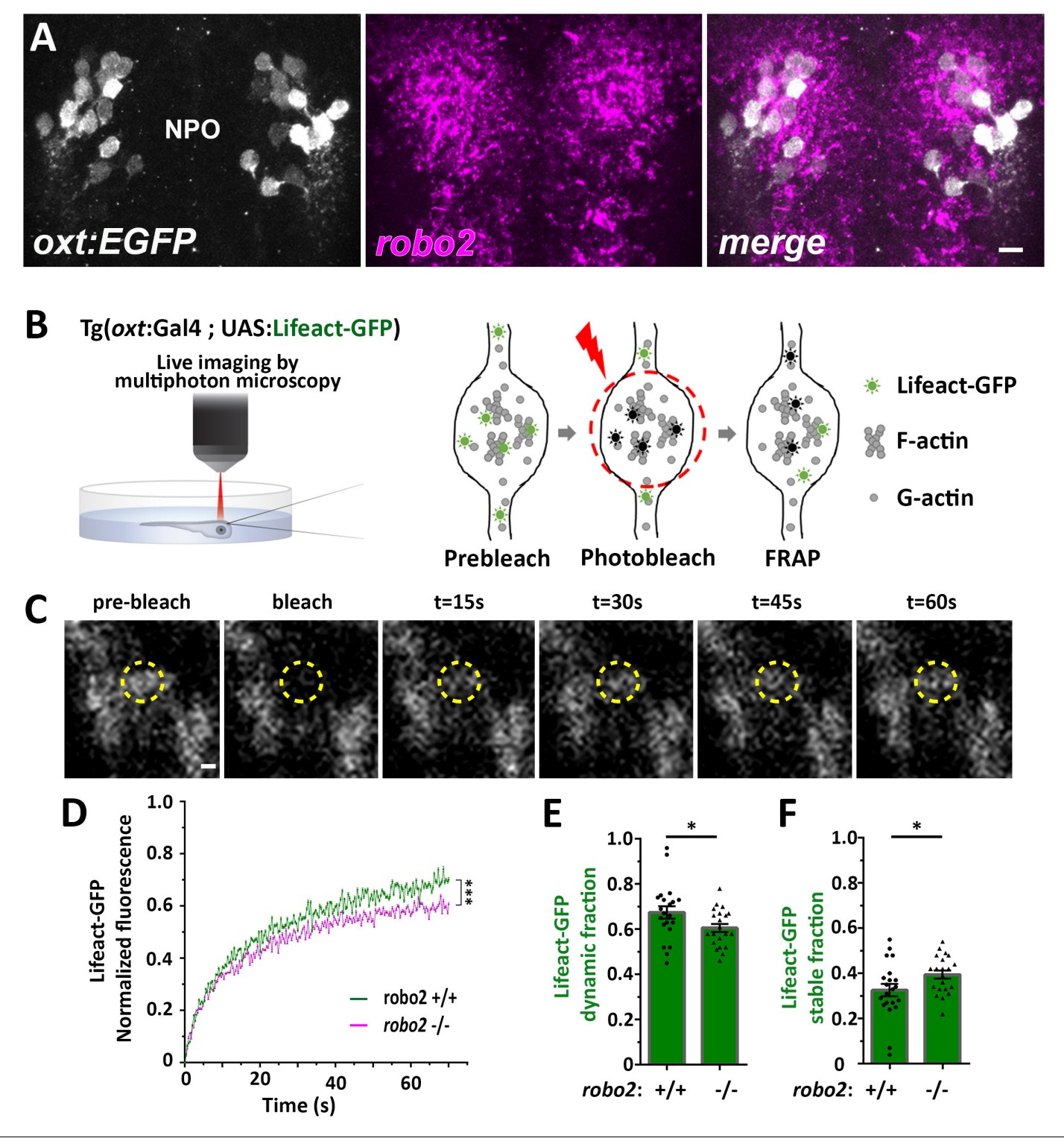

**Figure 4.** Robo2 regulates synaptic actin dynamics. (**A**) *robo2* is expressed in larval zebrafish neurosecretory preoptic area (NPO) and colocalizes with Oxytocin neurons. Confocal Z-stack images showing fluorescent in situ hybridization (FISH) of transgenic larvae Tg(*oxt*:EGFP) (3 days post-fertilization (dpf)) using probes directed against *robo2* mRNAs (magenta), followed by anti-EGFP staining. The neurosecretory preoptic (NPO) area in which OXT neurons were labeled is shown. Scale bar: 20 μm. (**B,C**) Real-time monitoring of synaptic actin dynamics in live transgenic reporter Tg(*oxt*:Gal4; UAS: Lifeact-EGFP) larvae mounted in 0.1% low-melt agarose and imaged using multi-photon microscopy upon Fluorescence Recovery after Photobleaching (FRAP) (**B**). Time-series images of FRAP experiment in a neurohypophyseal synapse with Lifeact-EGFP expression (**C**). Scale bar: 200 nm. (**D–F**)
*Figure 4 continued on next page*

Figure 4 continued

Assessment of synaptic actin dynamics in *robo2* mutant using the transgenic actin dynamics reporter Tg(*oxt*:Gal4; UAS:Lifeact-EGFP) larvae. Graph showing the normalized FRAP profile of Lifeact-EGFP fluorescence intensity in 6-dpf *robo2*+/+ (n = 19 synapses from seven larvae) and *robo2*-/- (n = 21 synapses from seven larvae) (D) (***p<2e-16) for genotypeXtime interaction effect in a linear mixed effects model to account for inter-synapse and inter-genotype variability (see Materials and methods). Bar graphs showing the dynamic (E) and stable (F) Lifeact-EGFP fractions in *robo2*$^{+/+}$ vs *robo2*$^{-/-}$ neurohypophyseal synapses (*p<0.05 Student's *t*-test; with Cohen's d = 0.67). Error bars indicate SEM in (D–F).
The online version of this article includes the following source data for figure 4:

**Source data 1.** Experimental data for *Figure 4*: Actin dynamics in neurohypophyseal synapses.

to study the role of Slit3 in regulating synaptic OXT content. Knock-down of *slit3* using a previously validated morpholino antisense oligonucleotide targeted to the translational start site (*Barresi et al., 2005*), led to reduced synaptic OXT content similar to the *robo2* mutant phenotype (*Figure 5F–J*). Thus, the volume of OXT puncta was significantly reduced following *slit3* knockdown, compared to mock injected controls (Cohen's d effect size 1.12, *p* value < 0.01), while the number, size of neurohypophyseal synapses and number of neurohypophyseal OXT axonal projections were unaffected. To investigate the role of Slit3 locally, we employed the *Tg(pomc:Gal4)* driver that enabled local secretion of Slit3-EmGFP fusion protein from proopiomelanocortin (POMC) cells that are arranged in two clusters, anterior and posterior, abutting neurohypophyseal OXT axons and synapses respectively (*Figure 5K*). We generated mosaic transgenic clones expressing Slit3-EmGFP in anterior and posterior hypophyseal locations (*Figure 5L*). Interestingly, when comparing to tRFP expressing clones we detected enlarged volume of OXT puncta adjacent to Slit3-EmGFP-positive clones located in a posterior but not anterior position (*Figure 5M–R*). As depicted schematically in *Figure 5L*, this localized effect of Slit3 coincided with the posterior position of the endogenous OXT synapses. These results suggest that localized context-dependent Slit3-Robo2 signaling regulates synaptic OXT levels.

## Robo2 regulates synaptic OXT vesicles accumulation

We next asked if Robo2 plays a role in synaptic accumulation of OXT-loaded vesicles. To visualize OXT neuropeptide dynamics in real time, we developed a novel transgenic tool, *Tg[oxt:OXTSP-EGFP-OXT-NP]*, in which OXT promoter drives the expression of EGFP fused with the OXT precursor, between the signal sequence and the OXT peptide (*Figure 6A*). Efficient production of a cleaved EGFP-OXT fusion protein was confirmed by Western blot analysis of pituitary synaptosomes isolated from adult pituitaries of transgenic fish (*Figure 6B*). The vesicular synaptic expression of the EGFP-OXT fusion protein was validated by triple co-immunostaining of the transgenic EGFP-tagged product, together with two specific antibodies to the endogenous neurophysin and OXT nonapeptide. This revealed that the EGFP-tagged OXT transgenic protein product mainly co-localized with the mature (i.e. cleaved and cyclised) endogenous OXT neuropeptide (*Figure 6C* and *Figure 6—figure supplement 1*). As expected, in some cases, the OXT-GFP reporter protein co-localized with vesicles containing both cleaved and OXT-neurophysin protein, suggesting that it reported an intermediate step in the proteolytic processing of the OXT precursor peptide (*Figure 6C*). To verify the vesicular synaptic localization of the EGFP-tagged OXT we performed super-resolution microscopy (STORM) imaging of *Tg[oxt:OXTSP-EGFP-OXT-NP]* larvae. Our results showed that similar to the endogenous OXT, the EGFP-OXT reporter exhibited a clustered organization, indicative of large dense core vesicular organization of OXT (*Figure 6D,E* and *Figure 6—video 1*). To detect F-actin co-localization with EGFP-OXT labeled vesicles in neurohypophyseal synapses, we used the Calponin domain of Utrophin fused to mCherry (UAS:mCherry-Utrophin-CH), which is a specific F-actin probe (*Melak et al., 2017*). We expressed mCherry-Utrophin-CH in OXT neurons in combination with our Tg(*oxt*:oxt-SP-EGFP) that labels OXT-loaded vesicles in synapses and found that mCherry-Utrophin-CH co-localizes with OXT-loaded synaptic vesicles (*Figure 6F*). This result further supports our original suggestion that that F-actin is enriched near OXT synaptic vesicles.

To assess the vesicular mobility and monitor in vivo neuropeptide homeostasis using our novel transgenic OXT vesicles reporter, we performed FRAP analysis of individual neurohypophyseal synapses using two-photon microscopy in live 6 days post-fertilization transgenic *Tg[oxt:OXTSP-EGFP-OXT-NP]* larvae (*Figure 6G*). Upon bleaching, we observed gradual recovery of EGFP-OXT fluorescence indicating the mobilization of transiting OXT-loaded vesicles in the synapses. The extent of

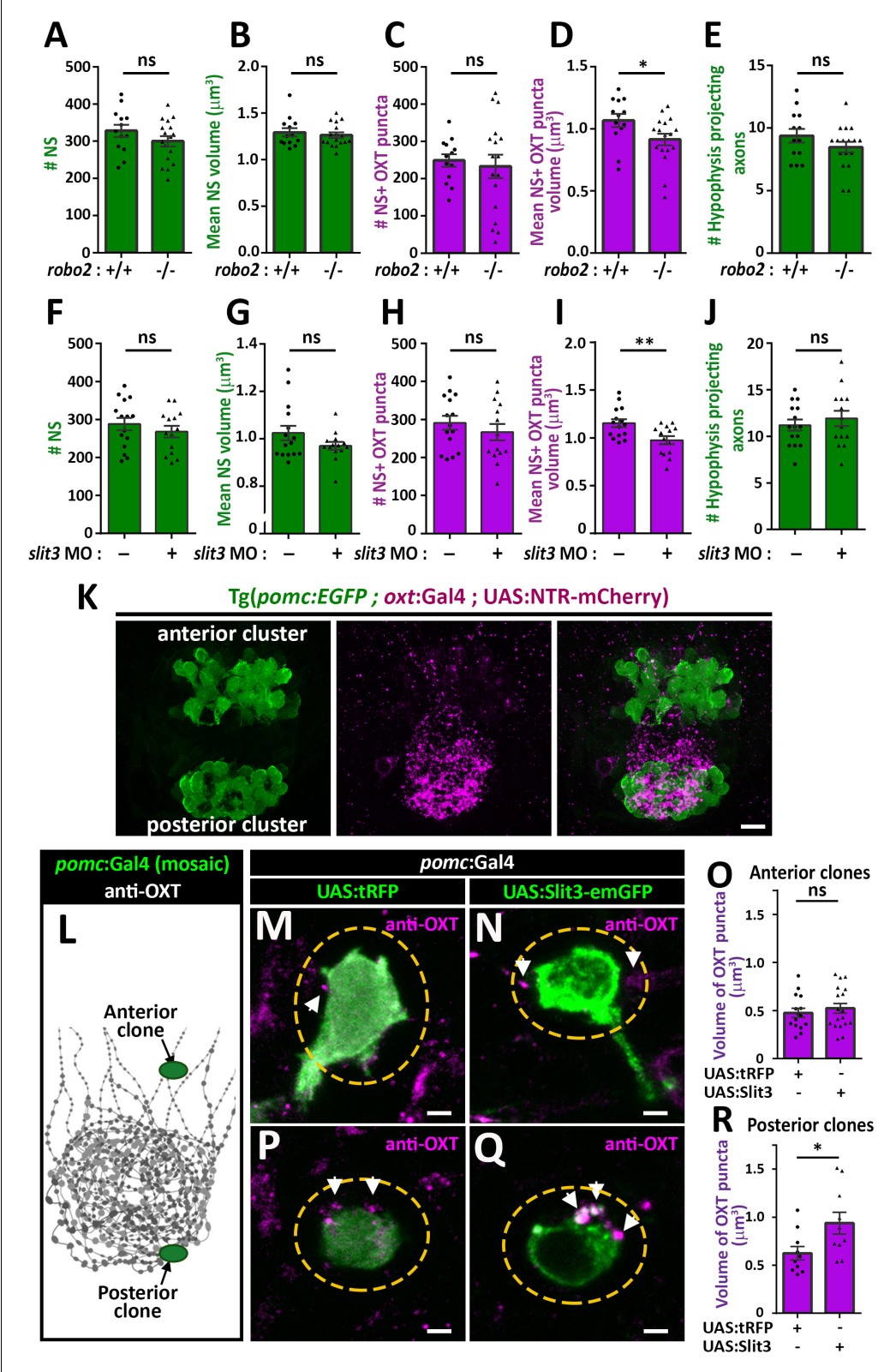

**Figure 5.** Slit3-Robo2 signaling regulates synaptic OXT levels. (A–E) Assessment of synaptic oxytocin content in *robo2* mutant was performed as described in *Figure 1F*. Graph showing the number and size of neurohypophyseal synapses (NS) (A,B) and colocalizing OXT puncta (C,D) and the number of neurohypophyseal projecting axons (E) in 8 days post-fertilization (dpf) *robo2+/+* (n = 13) vs *robo2-/-* (n = 17) larvae (*p<0.05 Student's *t*-test with Cohen's d = 0.82; ns denotes not significant). (F–J) Assessment of synaptic oxytocin content upon *slit3* knock-down. Transgenic (*oxt*:EGFP)

*Figure 5 continued on next page*

*Figure 5 continued*
embryos were injected with injection buffer or injection buffer with 0.85 ng of morpholino targeted to the translational start site of *slit3*. Graph showing the number and size of NS (F,G) and colocalizing OXT puncta (H,I) and the number of neurohypophyseal projecting axons (J) in 8 days post-fertilization (dpf) control (n = 15) vs slit3 morpholino injected (n = 14) larvae (**p<0.05 with Cohen's d = 1.23; ns denotes not significant, Student's *t*-test). (K) Hypophyseal POMC cells are localized near OXT synapses. Confocal Z-stacks maximum intensity projection of hypophysis region of 5 day old triple transgenic (*pomc*:EGFP; *oxt*:Gal4; UAS:NTR-mCherry). The hypophysis area showing OXT NS adjacent to anterior and posterior clusters of POMC cells are shown. Scale: 10 µM. (L–R) Local overexpression in mosaic hypophyseal POMC clones located. Transgenic embryos expressing the *pomc*:Gal4 driver were injected with transposon-based transgenic vectors containing either control UAS:tRFP (M,P) or UAS: Slit3-EmGFP (N,Q). The injected 5-dpf larvae were subjected to immunostaining with anti-OXT protein and anti-GFP. OXT-puncta (arrows) within a distance of 2 µM (yellow dashed ellipse) from the clone surface were quantified. Scale bar: 2 µm. Bar graphs showing the mean volume of OXT puncta upon expression of tRFP vs Slit3 in hypophyseal POMC anterior (n = 15 and 18, respectively) or posterior clone (n = 10 and 10, respectively) position (*p<0.05 Student's *t*-test with Cohen's d = 1.14). Error bars indicate SEM in (A-J, O and R).

The online version of this article includes the following source data for figure 5:

**Source data 1.** Experimental data for *Figure 5*: Neurohypophyseal synaptic parameters upon gain- or loss-of function of *slit3* or *robo2*.

fluorescence recovery was low (13%), suggesting that the majority of the bleached OXT-EGFP-positive vesicles were stationary and not mobile (*Figure 6H,I*). FRAP analysis in transgenic larvae on the background of *robo2* mutants revealed that the fluorescence recovery rate in *robo2*-/- mutants, was significantly lower than in *robo2*+/+ larvae (*Figure 6I*). Taken together, these results suggest that Robo2-mediated signaling regulates actin dynamics as well as the accumulation of OXT-containing vesicles in neurohypophyseal synapses.

## Robo2 and Cdc42 regulate OXT neuropeptide levels

We hypothesized that Robo2 exerts its effect on OXT content via actin polymerization; thus we searched for a candidate signaling mediator that could link between these two Robo2-mediated effects on actin dynamics and OXT content. Robo signaling is known to affect the transition of the Rho-GTPase protein Cdc42 from GTP- to GDP- bound state, resulting in decreased actin polymerization (*Wong et al., 2001*). We therefore tested if conditional OXT-specific expression of dominant-negative (i.e. GDP-bound) mutant form of CDC42, termed Cdc42(T17N), would affect synaptic OXT content and whether the effect would be dampened in a Robo2 mutant. We used our *Tg(oxt:Gal4)* transgenic fish to drive specific oxytocinergic expression of EGFP-Cdc42(T17N) fusion protein, which was regulated by ten Gal4 DNA binding UAS repeats (*Ando et al., 2013*). Thus, one cell-stage embryos were co-injected with transposon-based DNA constructs harboring *oxt*:Gal4 together with either UAS:EGFP-Cdc42-T17N or UAS:EGFP construct as a control. (*Figure 7A*). We then quantified the levels of OXT and EGFP-Cdc42(T17N) proteins in each synapse (*Figure 7B*). We took advantage of the variable expression of the injected construct in each individual synapse, to examine whether differences in OXT content correlated with expression levels of EGFP-Cdc42(T17N) or the control EGFP. Regression analysis of OXT fluorescence as a function of EGFP fluorescence in each injected zebrafish larvae showed that high EGFP fluorescence led to decreased OXT levels in both *robo2*+/+ and *robo2*-/- larvae (*Figure 7C*; p<0.01, adj. $R^2$ = 0.93 and 0.85 respectively) (*Figure 7C*). This effect is likely due to overexpression of the untethered EGFP. In contrast, expression levels of EGFP-Cdc42(T17N) were positively correlated with increased OXT content in *robo2*+/+ (*Figure 7D*; p<0.01, adj. $R^2$ = 0.74). However, the positive effect of EGFP-Cdc42(T17N) on synaptic OXT content was dampened in *robo2*-/- mutants (*Figure 7D*; p=0.35, Adj. $R^2$ = 0.52), a finding which is consistent with the fact that Robo2 inactivation leads to increased levels of active GTP-bound Cdc42, which may counteract Cdc42(T17N). In contrast to the aforementioned effect of dominant-negative EGFP-Cdc42(T17N) mutant protein, overexpression of EGFP-fused constitutively-active (i.e. GTP-bound) mutant form of CDC42, termed Cdc42(G12V), led to a dose-dependent decreased OXT levels in EGFP positive synapses similar to the low OXT levels in *robo2* -/- synapses (*Figure 7—figure supplement 1*). These results place the small GTPase Cdc42, which is a key regulator of actin dynamics, in a Robo2 signaling cascade that controls the levels synaptic OXT neuropeptide (*Figure 7E*).

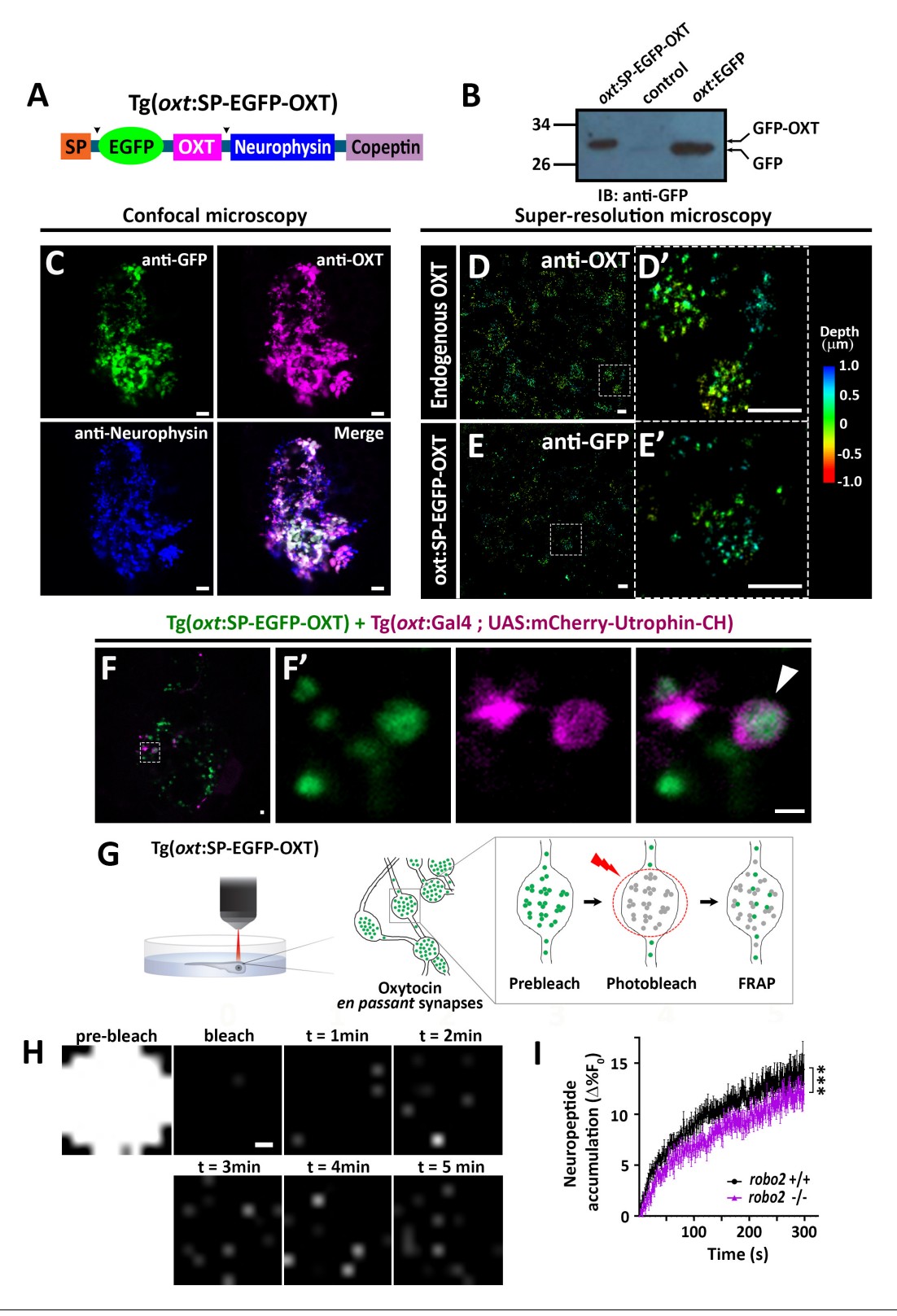

**Figure 6.** Robo2 regulates synaptic OXT dynamics. (**A**) Schematic of novel transgenic OXT tool Tg(*oxt*:OXT-SP-EGFP-OXT), in which *oxt* promoter drives expression of Oxytocin precursor protein with an internally-tagged EGFP at the C-terminus of the signal peptide. (**B**) Validation of OXT-fusion EGFP protein expression by Western blot analysis. Pituitary protein extracts from adult Tg(*oxt*:OXT-SP-EGFP-OXT), TL (control) and Tg(*oxt*:EGFP) zebrafish were immunoblotted using anti-GFP. (**C**) Validation of OXT-fusion EGFP protein expression by immunohistochemistry. Confocal Z-stack

*Figure 6 continued on next page*

*Figure 6 continued*

images of the neurohypophyseal area of 5-dpf transgenic Tg(*oxt*:OXT-SP-EGFP-OXT) larvae. Larvae were stained using anti-GFP, anti-OXT and anti-neurophysin antibodies. Scale bar: 5 μm. (D–E) Assessment of endogenous vs OXT-fusion EGFP protein localization and packaging by super-resolution microscopy. STORM Z-stack images of the endogenous synaptic OXT (D) in comparison to OXT-EGFP fusion in transgenic Tg(*oxt*:OXT-SP-EGFP-OXT) larvae (E). five dpf larvae were stained with either anti-OXT (D) or anti-GFP (E) antibodies and visualized by STORM as described in the Method section. The right panels D' and E' show magnifications of the represented areas outlined in D) and E) Scale bars: 1 μm. Color code indicates Z-axis depth. (F) Spatial relationship between F-actin and OXT-fusion EGFP neuropeptide in hypophysis. Tg(*oxt*:OXT-SP-EGFP-OXT) embryos expressing the OXT-fusion were injected with transposon-based transgenic vectors Tg(oxt:Gal4) and Tg(UAS:mCherry-Utrophin-CH) expression cassettes. Confocal Z-stack images of the neurohypophyseal area of 5-dpf transgenic larvae (F). Maginification showing the EGFP-positive synapses and Utrophin-labeled mCherry-positive F-actin in the synapse (F'). Scale bars: 1 μm. (G,H) Real-time monitoring of synaptic oxytocin vesicle dynamics in live transgenic reporter Tg(*oxt*:OXT-SP-EGFP-OXT) larvae. Schemata of the experimental design (G): six dpf larvae were mounted dorsally in low-melt agarose gel, submerged in E3 embryo buffer and imaged using multi-photon microscopy. Neurohypophyseal synapses containing OXT-EGFP vesicles were photobleached and the fluorescence recovery after photo bleaching (FRAP) over time was monitored. The fluorescent recovery occurs due to the dynamic exchange of mobile and unbleached OXT-EGFP vesicles from neighboring *en passant* synapses over time (t = 1–5 min). The brightness and contrast of the images were increased to visualize the individual pixels (H). Scale bar: 200 nm. (I) Graph showing neuropeptide accumulation (normalized FRAP curves) of OXT-EGFP in neurohypophyseal synapses in 6-dpf *robo2*+/+ (n = 9 larvae) vs robo2-/- (n = 6 larvae) (***p<2.76e-05) for genotypeXtime interaction effect in a linear mixed effects model to account for inter-synapse and inter-genotype variability (see Materials and methods). Neuropeptide accumulation represents full scale normalized data to account for differences in synaptic OXT-EGFP fluorescence, that is fluorescent values upon photobleaching were normalized to zero. Error bars indicate SEM.

The online version of this article includes the following video, source data, and figure supplement(s) for figure 6:

**Source data 1.** Experimental data for *Figure 6*: OXT-EGFP dynamics in neurohypophyseal synapses.

**Figure supplement 1.** Specificity of antibodies to oxytocin and neurophysin.

**Figure 6—video 1.** Three-dimensional render showing STORM images of endogenous OXT in neurohypophyseal synapses of 5 days post-fertilization (dpf) old zebrafish larvae.

https://elifesciences.org/articles/45650#fig6video1

## Discussion

To maintain body homeostasis, neurons sense and integrate a multitude of environmental and physiological signals and evoke a response when a deviation is detected. To achieve this, neurosecretory synapses of these cells must maintain adequate levels of neuropeptides that are readily primed to be secreted. This is particularly relevant for neuroendocrine signals that occur at the neurohypophysis level where two neuropeptides that are essential for homeostasis, oxytocin and vasopressin are released into the blood circulation to exert their effects on peripheral organs (*Miyata, 2017*; *Wircer et al., 2016*). Thus, how neuropeptides homeostasis is maintained in such synapses is a fundamental question.

Recent studies reported that synaptic F-actin regulates multiple aspects of vesicular homeostasis ranging from vesicle capture to release and recycling (*Chia et al., 2014*; *Ganguly et al., 2015*; *Guillet et al., 2016*; *Marra et al., 2012*; *Miki et al., 2016*; *Soykan et al., 2017*; *Stavoe and Colón-Ramos, 2012*; *Vincent et al., 2015*). Actin is also required for recruitment of multiple synaptic proteins and receptors that are essential for synaptic function (*Sankaranarayanan et al., 2003*). Here, we focused on the neuropeptide OXT, which is stored in LDCVs in numerous axonal swellings. These swellings act as *en passant* synapses, which are highly enriched in neurohypophyseal axonal projections (*Morris and Pow, 1988*). We found that local actin dynamics in those synapses regulate the levels of OXT neuropeptide and identified a new signal transduction machinery, Slit3-Robo2-Cdc42, which regulates synaptic actin dynamics to maintain steady-state OXT content in those synapses.

The majority of reported studies on synaptic OXT content and release were performed on organotypic cultures or by electron microscopy on sliced mammalian neurohypophyseal tissues (*Alonso et al., 1981*; *Miyata et al., 2001*; *Tobin and Ludwig, 2007*). Using TEM we demonstrated that as in mammals the neurohypophysis of larval zebrafish contains classical neurosecretory synapses that are enriched with electron-dense LDCVs and observed vesicle secretion and recycling events. Consistent with the synaptic identity of OXT-positive neurohypophyseal axonal swellings, nearly all analyzed oxytocin-positive puncta co-localize with a synaptic vesicle reporter Synaptophysin-EGFP. Monitoring synaptic OXT content in the transparent zebrafish larvae, we were able to study the regulation of OXT neuropeptide levels in the context of a living vertebrate animal without invasive manipulation. We demonstrated the robustness and validity of our experimental system by

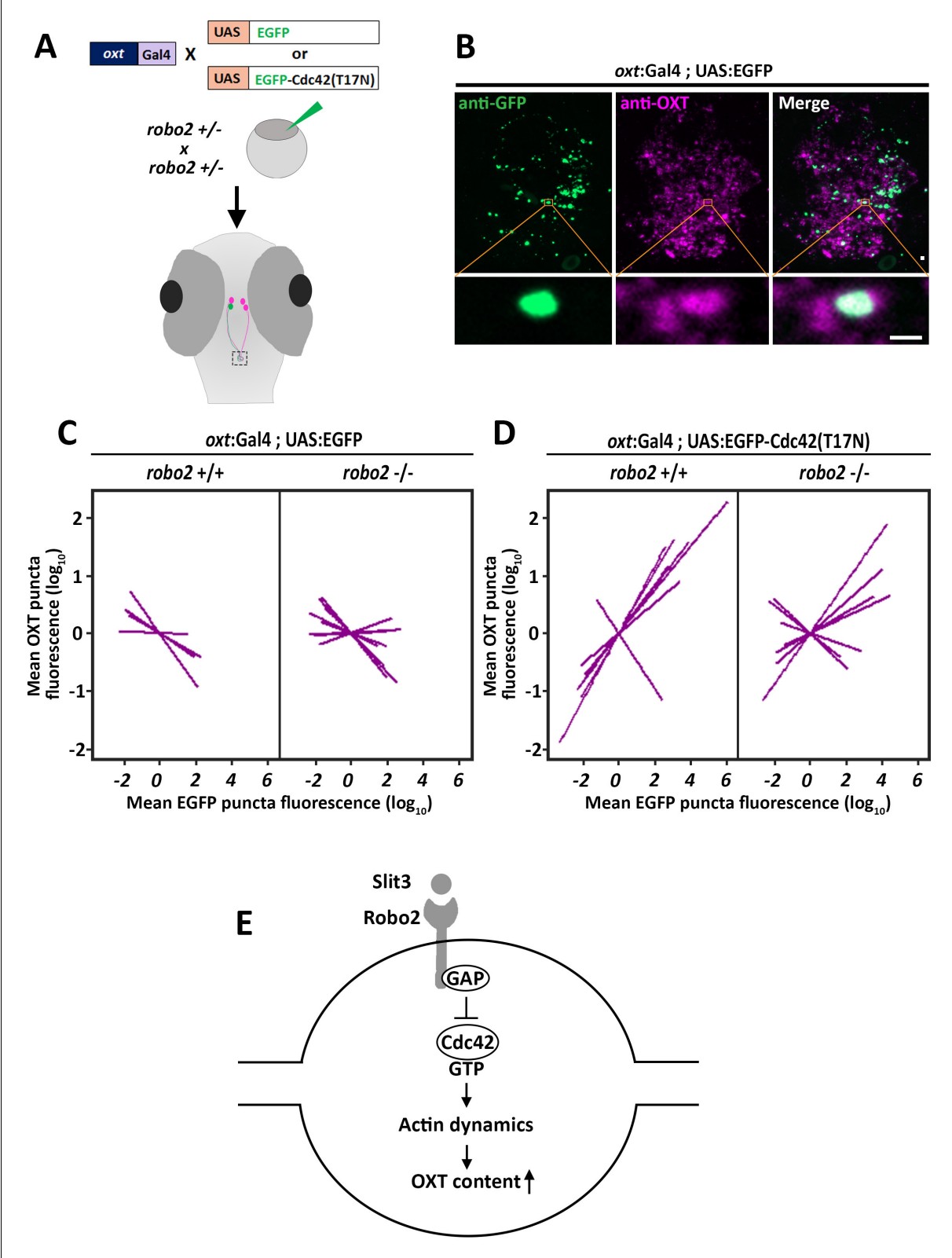

**Figure 7.** Robo2 regulates synaptic OXT levels via Cdc42. (A,B) Assessment of the effect of Oxytocin neuron-specific overexpression of actin-regulating protein Cdc42 in Tg(*oxt*:Gal4) larvae. Transgenic embryos expressing the *oxt*:Gal4 driver were injected with transposon-based transgenic vectors containing either control UAS:EGFP or UAS:Cdc42(T17N)-EGFP. Larvae were fixed at 8 days post-fertilization (dpf) and immunostained with anti-GFP and anti-OXT antibodies and neurohypophyseal synapse (NS) were identified as described earlier. Maximum intensity projection (MIP) reveals mosaic

*Figure 7 continued on next page*

*Figure 7 continued*

labeling of hypophyseal projecting axons (**B**). Bottom panel show magnifications of squared areas showing single stack with colocalization of axonal swelling with OXT puncta. Scale bar 1 μm. (**C,D**) Linear regression analysis comparing between synaptic EGFP levels (mean EGFP puncta) as a function of OXT fluorescence (mean OXT puncta). The data were normalized using mean-centering approach using the scale function in the 'R' software. Each line represents a regression line of single animal. The correlation between the mean ($\log_{10}$) fluorescence value of OXT puncta and GFP expression in neurohypophyseal synapse was tested with a linear regression model (ANCOVA), accounting for the effect of GFP fluorescence, and individual fish. Mean EGFP fluorescence is inversely correlated with OXT fluorescence in *robo2+/+* (n = 5; p<0.01, adj. $R^2$ = 0.93) and in *robo2-/-* (n = 8; and p<0.01, adj. $R^2$ = 0.85) (**C**). Mean EGFP-Cdc42(T17N) fluorescence positively correlates with OXT fluorescence in *robo2+/+* (n = 7; p<0.01, adj. $R^2$ = 0.74) larvae but not in *robo2-/-* (n = 7; p=0.35, Adj. $R^2$ = 0.52) (**D**). (**E**) Model of the role of Robo2 in neurohypophyseal synapses: Synaptic Slit3-Robo signaling inactivates Cdc42 via GAP by affecting the transition of the Rho-GTPase protein Cdc42 from GTP- to GDP- bound state. Cdc42 inactivation reduces actin polymerization and increases synaptic OXT content.

The online version of this article includes the following source data and figure supplement(s) for figure 7:

**Source data 1.** Experimental data for *Figure 7*: Synaptic fluorescence values upon OXT-neuron specific Cdc42 over-expression.
**Figure supplement 1.** Cdc42 regulates synaptic OXT levels.

showing that synaptic OXT content is depleted upon osmotic challenge, which is in agreement with previous studies done in mammals and also in teleost fish (*Alonso et al., 1981*; *Balment et al., 1980*; *Neumann et al., 1993*; *Pierson et al., 1995*). Conversely, expression of the light chain of botulinum toxin serotype B specifically in the OXT neurons, led to increased synaptic OXT content. This is in agreement with the fact that neurohypophyseal neurons exhibit spontaneous activity and activity-dependent SNARE-mediated synaptic release, which is blocked by botulinum toxin (*Jurgutis et al., 1996*; *Tobin et al., 2012*).

It was reported that dual pools of actin filaments exist in synapses, a cytoplasmic pool associated with the vesicles and a second pool of cortical filaments associated with the plasma membrane (*Alonso et al., 1981*; *Bleckert et al., 2012*; *Nelson et al., 2013*). Using super-resolution STORM microscopy of OXT synapses in the Lifeact-EGFP reporter, we observed that actin filaments form cage-like structures that surround OXT-containing vesicles similar to reports in rat neurohypophyseal axonal termini (*Alonso et al., 1981*). The interaction of F-actin with OXT was further reinforced by co-localization of mCherry-Utrophin-CH with OXT-loaded synaptic vesicles. As neuropeptidergic (e.g. OXT) synapses lack active zone, which is typical in small transmitters synapses, the actin cage may regulate vesicle movement and exocytosis. Indeed, synaptic OXT content was decreased upon treatment of zebrafish larvae with cytochalasin D, a cell permeable mycotoxin known to inhibit F-actin formation (*Cooper, 1987*; *Goddette and Frieden, 1986*; *Lin et al., 1980*).

Synaptic actin filaments are dynamic structures which undergo treadmilling and regulate synaptic morphogenesis (*Bosch and Hayashi, 2012*; *Honkura et al., 2008*). Using FRAP analysis of the actin probe Lifeact-EGFP in OXT synapses of live animals, we found that synaptic Lifeact-EGFP dynamics is perturbed in *robo2* mutants. Thus, *robo2* mutants exhibit reduced mobile Lifeact-EGFP fraction and increased stable fraction, which was correlated to reduced OXT content in those synapses. It should be noted that using Lifeact-EGFP has several drawbacks including binding monomeric G-actin and competition with other F-actin binding actin regulators (e.g. Cofilin). Further experiments using FRAP analysis of Actin-EGFP, should be used to support our interpretation that the dynamics of synaptic polymerized actin is altered in *robo2* mutants. Having said that, a direct requirement of actin polymerization in OXT synapses was further supported by our finding that synaptic OXT content was reduced upon expression of actin depolymerizing protein Cofilin (*Shekhar and Carlier, 2017*; *Wioland et al., 2017*), specifically in the OXT neurons. Notably, it was reported that Cofilin affects synaptic plasticity and morphology (*Jang et al., 2005*; *Liu et al., 2016*; *Piccioli and Littleton, 2014*; *Pontrello et al., 2012*; *Zimmermann et al., 2015*) however, we did not observe any gross changes in the number or size of the OXT synapses, indicative of the specificity of our genetic manipulations. As cofilin-dependent actin dynamics has been reported to function in synaptic vesicle mobilization and exocytosis (*Wolf et al., 2015*), it may play a similar role in OXT synapses.

We recently identified that *slit3*, a cognate ligand for Robo receptors is highly expressed in pituicytes, the astroglial component of the neurohypophysis directly contacting OXT termini (*Anbalagan et al., 2018*). In addition to axonal guidance, Robo-Slit signaling is also known to regulate synaptogenesis and actin dynamics in other CNS regions (*Blockus and Chédotal, 2016*;

*Campbell et al., 2007*; *Kidd et al., 1998*; *Slováková et al., 2012*). However, the role of Robo-Slit3 in OXT synapse formation and function is not known. We show here that Robo2 is expressed in OXT neurons and genetic perturbations of both *slit3* and *robo2* affect synaptic OXT content. Notably, previous studies have shown that Slit-Robo signaling regulates actin dynamics during neuronal migration and axonal guidance (*Kidd et al., 1998*; *Slováková et al., 2012*; *Wong et al., 2001*). However, the reduction in synaptic OXT content, which we observed in *robo2 -/-* mutants and following *slit3* knockdown were not accompanied by axon guidance defects as the number of neurohypophyseal projecting axons were unaffected.

In view of our findings regarding the involvement of Robo2 in synaptic actin dynamics and OXT content, we hypothesized that changes in synaptic actin mobility regulate the steady-state accumulation of OXT-containing vesicles at the neurohypophyseal release site. To directly test this hypothesis, we established a novel transgenic zebrafish OXT-EGFP fusion line, allowing real time in vivo monitoring of the dynamicity of OXT-loaded vesicles. Notably, transgenic rats harboring a similar OXT-EGFP fusion have already been reported (*Hashimoto et al., 2014*); however, the deep anatomical location of the neurohypophysis makes it difficult to study the dynamic sub-cellular processes in mammalian species in vivo. Using our zebrafish transgenic reporter, we observed that the majority of the vesicles were immobile, as only 13% of non-bleached fluorescent vesicles exchanged into the photobleached synapses. This finding is in agreement with previous work on LDCV mobility in Drosophila neuroendocrine termini (*Bulgari et al., 2014*). We revealed that Robo2 regulates the accumulation of transiting OXT-loaded vesicles in the synapses. Whether this phenotype is due to decreased vesicle capture or increased exocytosis needs to be further investigated. As Robo2 and Slit3 are also involved in inhibition of dopamine secretion in the midbrain and insulin secretion in pancreatic cells (*Gore et al., 2017*; *Yang et al., 2013*), they might also function to inhibit synaptic OXT release and associated excitation-secretion coupling observed in these neurons (*Leng et al., 2008*).

To link synaptic actin dynamics to Robo2 signaling and regulation of neuropeptide content we targeted the intercellular GTPase Cdc42, which acts downstream of Robo signaling pathway to affect actin dynamics. During neuronal migration, Robo signaling promotes Cdc42-GTP hydrolysis that, in turn, attenuates actin polymerization (*Wong et al., 2001*). We show here that expression of the dominant-negative (i.e. GDP-bound) Cdc42(T17N) in OXT synapses led to increased OXT levels in WT fish in a dose-dependent manner while expression of a constitutively-active (i.e. GTP-bound) mutant form of CDC42 had a reciprocal effect. *robo2* mutant was somewhat refractory to the effect of Cdc42(T17N) on synaptic OXT levels, suggesting that Robo2 acts upstream of Cdc42 in the context of synaptic OXT content. These results are consistent with the notion that lack of Robo2 signaling leads to increased levels of GTP-bound Cdc42 (*Wong et al., 2001*) and reduced synaptic OXT levels (this study). Our findings regarding the regulation of synaptic OXT content might be relevant to other neuropeptidergic and also endocrine cells. Thus, interaction between Cdc42 and vesicle-associated membrane protein 2 (VAMP2) was previously shown in the case of insulin granules in pancreatic beta cells (*Nevins and Thurmond, 2005*). These authors suggested that a mechanism whereby glucose activates Cdc42 to induce the targeting of intracellular Cdc42-VAMP2-insulin vesicles to SNARE proteins at the plasma membrane. Notably, OXT, Robo2 and Slit3 have all been implicated in Autism spectrum disorders (*Anitha et al., 2008*; *Cukier et al., 2014*; *LoParo and Waldman, 2015*; *Modahl et al., 1998*; *Uzefovsky et al., 2019*; *Zhang et al., 2016*). Given the nature of neuroendocrine signaling, whereby relatively large quantities of neuropeptides are released in a highly coordinated and temporally controlled manner, it stands to reason that maintaining proper peptide levels at the synapse is crucial for the proper functioning of this system. Thus, further work is required to identify the exact OXT-related neuroendocrine and physiological consequences of impaired Slit3-Robo2-Cdc42 signaling pathway.

Taken together, our findings reveal that Slit3-Robo2-Cdc42 signaling modulates synaptic actin dynamics to maintain steady-state levels of OXT neuropeptide readily primed to be secreted upon physiological demand.

## Materials and methods

**Key resources table**

*Continued on next page*

*Continued*

| Reagent type (species) or resource | Designation | Source or reference | Identifiers | Additional information |
|---|---|---|---|---|
| Reagent type (species) or resource | Designation | Source or reference | Identifiers | Additional information |
| Genetic reagent, TL (D. rerio) | Tg(*oxt*:EGFP) wz01 | (*Blechman et al., 2011*) | ZDB-ALT-111103–1 | |
| Genetic reagent, TL (D. rerio) | Tg(*oxt*:gal4) wz06 | (*Anbalagan et al., 2018*) | ZDB-ALT-171113–2 | |
| Genetic reagent, TL (D. rerio) | Tg(*oxt*:OXTSP-EGFP-OXT)wz14 | This manuscript | ZDB-ALT-181219–3 | |
| Genetic reagent, TL (D. rerio) | Tg(*UAS*:sypb-EGFP)biu5 | (*Zada et al., 2014*) | ZDB-ALT-150115–1 | Lior Appelbaum (Bar-Ilan Univ.) |
| Genetic reagent, TL (D. rerio) | Tg(UAS:Lifeact-GFP)mu271 | (*Helker et al., 2013*) | ZDB-ALT-130624–2 | Wiebke Herzog (Univ. of Muenster) |
| Genetic reagent, TL (D. rerio) | Tg(UAS:BotxLCB-GFP)icm21 | (*Sternberg et al., 2016*) | ZDB-ALT-160119–9 | Claire Wyart (ICM, Paris) |
| Genetic reagent, TL (D. rerio) | Tg(UAS:NTR-mCherry)c264 | (*Davison et al., 2007*) | ZDB-ALT-070316–1 | |
| Genetic reagent, TL (D. rerio) | Tg(−1.0pomca:Gal4-VP16)wz05 | (*Anbalagan et al., 2018*) | ZDB-ALT-171113–1 | |
| Genetic reagent, TL (D. rerio) | robo2^ti272z | (*Fricke et al., 2001*) | ZDB-ALT-980203–1097 | Joshua Bonkowsky (Univ. of Utah) |
| Recombinant DNA reagent | Tol2 pDEST myl7:mCherry | (*Golan et al., 2016*) | | Berta Levavi-Sivan, HUJI, Rehovot |
| Recombinant DNA reagent | Tol2 10xUAS: EGFP; myl7: mCherry | This manuscript | | Materials and methods - Transgenesis experiments |
| Recombinant DNA reagent | Tol2 *oxt*:OXTSP-EGFP-OXT; *myl7*:EGFP | This manuscript | | Materials and methods - Transgenesis experiments |
| Recombinant DNA reagent | Tol2 UAS:Cdc42-G12V-EGFP | (*Ando et al., 2013*) | | Naoki Mochizuki (NCVC, Osaka) |
| Recombinant DNA reagent | Tol2 UAS:Cdc42-T17N-EGFP | (*Ando et al., 2013*) | | Naoki Mochizuki (NCVC, Osaka) |
| Recombinant DNA reagent | Tol2 pME mCherry-Utrophin-CH | (*Andersen et al., 2011*) | | Mary Hallaron (Univ. of Wisconsin) |
| Recombinant DNA reagent | Tol2 10xUAS: mCherry-Utrophin-CH; myl7:mCherry | This manuscript | | Materials and methods - Visualization of synaptic F-actin |
| Recombinant DNA reagent | Tol2 pME Slit3(SignalPeptide)-EmeraldGFP-Slit3 | This manuscript | | Joshua Bonkowsky (Univ. of Utah) |

*Continued on next page*

*Continued*

| Reagent type (species) or resource | Designation | Source or reference | Identifiers | Additional information |
|---|---|---|---|---|
| Recombinant DNA reagent | Tol2 10xUAS: Slit3(SignalPeptide)-EmeraldGFP-Slit3; myl7:mCherry | This manuscript | | Materials and methods - Transgenesis experiments |
| Antibody | Guinea pig polyclonal, anti-OXT | Peninsula labs | T-5021; RRID: AB_518526 | (1:200) |
| Antibody | Rabbit polyclonal, anti-GFP | ThermoFisher | A11122; RRID: AB_221569 | (1:200) |
| Antibody | Mouse monoclonal, anti-Neurophysin | (*Ben-Barak et al., 1984*) | PS45; RRID: AB_2062089 | Harold Gainer (NINDS, Bethesda) |
| Antibody | Alexa 488- or 647- Secondary antibodies | Jackson ImmunoResearch Laboratories | | (1:200) Materials and methods - Immuno-fluorescent staining |
| Antibody | Alexa 568- or 647- Secondary antibodies | Invitrogen | | (1:2000) Materials and methods - Visualization of synaptic F-actin |
| Chemical compound | Cytochalasin D | Sigma | C8273 | 400 nM |
| Chemical compound | Glucose oxidase | Sigma | G2133 | 8440 AU |
| Chemical compound | Catalase | Sigma | C40 | 70200 AU |
| Software | R | (*R Development Core Team, 2013*) | | Materials and methods – Statistical analysis |
| Software | Turboreg plugin | (*Thévenaz et al., 1998*) | | Materials and methods – Statistical analysis |
| Software | EasyFRAP | (*Rapsomaniki et al., 2012*) | | Materials and methods – FRAP analysis |
| Sequence-based reagent | TATATCCTCTGA GGCTGATAGCAGC | Gene Tools, (*Barresi et al., 2005*) | ZDB-MRPHLNO-050927–3 | Materials and methods - Transgenesis experiments. *slit3* knockdown |
| Sequence-based reagent | gaatgactcctcgtcgctct and gctgaggcatcttgtctgta | Sigma | | Materials and methods -Animals. *robo2* genotyping |
| Sequence-based reagent | gcatttacaacagctccatc | Sigma | | Materials and methods -Animals. *robo2* sequencing primer |
| Sequence-based reagent | tgtacaggcagatgtcaggc and TAATACGACTC ACTATAGGG-tcctcctccagtagagccag | Sigma | | Materials and methods - In situ hybridization. PCR primers for *robo2* probe |

## Animals

Zebrafish were raised and bred according to standard protocols. All experimental procedures were approved by the Weizmann Institute's Institutional Animal Care and Use Committee (IACUC). Animals were genotyped by Sanger sequencing. Transgenic zebrafish lines Tg(*oxt*:EGFP)[wz01] (*Blechman et al., 2011*), Tg(*oxt*:gal4)[wz06] (*Anbalagan et al., 2018*), Tg(*UAS*:Lifeact-EGFP)[mu271]

(*Helker et al., 2013*), Tg(*UAS*:BotxLCB-EGFP)[icm21] (*Sternberg et al., 2016*), Tg(UAS:NTR-mCherry)[c264] (*Davison et al., 2007*) and robo2[ti272z] (*Fricke et al., 2001*) were used in this study.

## Transgenesis experiments

We used the Tol2kit transposon-based transgenic vector system for site-specific recombination-based cloning (*Kwan et al., 2007*) and generated all plasmid DNA constructs (See Key Resources Table for list of recombinant DNA reagents). Briefly, genes were cloned into middle entry Tol2 plasmid 218. The DNA sequence of the cloned genes were checked for integrity by DNA Sanger sequencing. The resulting resulting plasmid was cloned downstream of 10xUAS promoter in Tol2 pDEST 395 plasmid with selection marker (green or red heart).

To generate OXTSP-EGFP-OXT construct, EGFP was cloned downstream of signal peptide sequence of zebrafish *oxt* gene (DNA sequence coding for first 20 amino-acids) by overlap extension PCR and cloned into middle entry Tol2 plasmid 218. The resulting plasmid was recombined downstream of *oxt* promoter in Tol2 pDEST 395 plasmid. The resulting plasmid was coinjected with Tol2 transposase mRNA and founders were outcrossed to obtain germline transmitting lines Tg(*oxt*:OXTSP-EGFP-OXT; *myl7*:EGFP)[wz14].

For *slit3* knockdown by morpholino, morpholino oligonucleotides (MOs) (Gene Tools) targeted to the translation start site of *slit3* were used as described previously (*Barresi et al., 2005*). MO stock was prepared by dissolving it in distilled water at 1 mM and embryos at one-cell stage were micro-injected with ATG-morpholino (0.85 ng/embryo) or mock-injected and allowed to develop at 28.5°C.

For local expression of *slit3*, we performed mosaic transgenesis in POMC+ cells adjacent to axons. Tg(*pomc*:Gal4) embryos at one-cell stage were micro-injected with vector *pomc:Gal4* in combination with 10xUAS:tRFP-caax or 10xUAS:Slit3-emGFP. All the plasmids were injected at concentration of 20 ng/µl each and with transposase mRNA at a concentration of 20 ng/µl (~500 pl/embryo). Using this method, we were able to attain cell labeling in ~10% of the surviving embryos. 5-dpf larvae that expressed heart marker were collected and fixed in 4% PFA. Due to the mosaic nature of the transgenesis experiments, and sparse labeling of POMC cells in the hypophysis, transgenesis experiments were performed individually.

## Hyperosmotic challenge

8 days post-fertilization (dpf) transgenic Tg(*oxt*:EGFP)[wz01] or Tg(*oxt*:Gal4; UAS:BotxLCB-EGFP; UAS:NTR-mCherry) larvae were treated with 25% artificial sea water (1.75 g Instant ocean Sea salt in 200 mL Danieau buffer) for a period of 60 min and then washed and returned to Danieau's medium for additional 60 min. The larvae were incubated at 28°C during entire procedure and larvae were fixed in 4% PFA overnight at 4°C prior to immunostaining.

## Actin perturbation

For temporal inhibition of actin filaments, a 100 µM stock of cytochalasin D (Sigma C8273) was prepared in DMSO. Briefly, 15–20 Tg(*oxt*:EGFP) 4-dpf larvae were treated with cytochalasin D (400 nM) for 24 hr in a 12-well plate at 28°C. Control embryos were treated with equivalent concentrations of DMSO.

For specific perturbation of actin in oxytocin neurons, Tg(*oxt*:gal4) embryos were micro-injected with vector *oxt*:gal4 in combination with 10xUAS:EGFP or 10xUAS:cofilin-EGFP or 10xUAS:*cdc42*(T17N)-EGFP or 10xUAS:cdc42(G12V)-EGFP (*Ando et al., 2013*). All the plasmids were injected at a concentration of 20 ng/µl each and with transposase mRNA at a concentration of 20 ng/µl (~500 pl/embryo). Using this method, we were able to attain cell labeling in ~10% of the surviving embryos. 8-dpf larvae that expressed the additionally expressed EGFP heart marker were sorted and fixed in 4% PFA prior to immunostaining. Due to the mosaic nature of the transgenesis experiments and sparse labeling of OXT neurons projecting to the hypophysis (<5% of mosaic clones), transgenesis experiments were performed separately for each constructs.

## In situ hybridization

RNA in situ hybridization was performed as described in *Machluf and Levkowitz (2011)*. For *robo2* probe synthesis, partial coding sequences and 3' UTR of the genes were amplified by PCR, along with a T7 tail in the reverse primer, and purified with PCR cleanup kit. The purified products served

as a template to synthesize digoxigenin-labeled antisense mRNA probes using DIG RNA labeling mix (Roche #11277073910).

## Immunofluorescent staining

For immunofluorescent staining, PFA-fixed larvae were washed in PBS (2 × 10 min), dehydrated using methanol series (25-50-75–100%) and stored at −20°C overnight. The samples were rehydrated from methanol (75-50–25%) to PBS, washed in PBS-Tx (Triton X100, 0.3%; 2 × 10 min) and blocked in 500 μL of blocking solution (PBS + 10% goat serum +1% DMSO+0.3% Triton X100) for 30 min at room temperature. The solution was then replaced with 200 μL of fresh blocking solution with primary antibodies at 1:200 concentration and incubated overnight at 4°C. Samples were washed with PBS-Tx (3 × 30 min) and treated with 200 μL of secondary antibodies in blocking solution at 1:200 concentration, overnight at 4°C. Then, samples were washed with PBS-Tx (3 × 30 min), transferred to 75% glycerol (25-50–75%) and the jaws were removed using a pair of hypodermic syringe before mounting the larvae with it's ventral side facing the objective. Rabbit anti-EGFP (ThermoFisher A11122), Guinea Pig anti-OXT (Peninsula labs T-5021) and secondary antibodies were obtained from Jackson ImmunoResearch Laboratories (West Grove, PA).

## Confocal imaging and image analysis

Samples were imaged by using Zeiss LSM 710 or LSM800 confocal microscopes with oil immersion 40X objective. Maximum intensity projection (MIP) images of the whole Z-stacks or subset of Z-stacks were generated using the Zen software (Zeiss). Processing of multiple channel images (i.e., linear adjustments of brightness, contrast and levels) was performed on individual channels using Photoshop CS7 Extended (Adobe). Images were analysed using the open source Fiji image-processing package and Volocity (Perkin Elmer). The number of hypophyseal synapses and OXT puncta was quantified using object measurement tool in Volocity (PerkinElmer) object identifier tool (thresholding was based on SD of fluorescence >= 4; size of 0.2 to 25 μm$^3$). The data were extracted and analysed using R (*R Development Core Team, 2013*).

For calculating colocalization coefficient of Syp-EGFP and OXT puncta fluorescence, all the quantitative measurements were conducted on the ZEN black software colocalization module. For each of the scanned larvae (n = 5) an ROI encapsulating the entirety of the hypophysis was manually delineated for each animal. Thresholds were kept at a constant value for all samples, that eliminated most background signal without losing any significant signal originating from stained compartments (17 for the cy5 channel, 15 for the 488 channel). Following this, weighted colocalization coefficients for each channel (M1 and M2), was calculated for each slice in which the hypophyseal synapses was visible (9–10 slices per animal). These values were averaged within each animal to obtain a total mean hypophysis colocalization index for that animal.

## Fluorescence recovery after photobleaching (FRAP) and image analysis

For live imaging of synaptic Lifeact-EGFP dynamics, 6-dpf live embryos were mounted dorsally in 0.1% low-melt agarose in a 12 mL plate and immersed in Danieau buffer with 0.3% tricaine to prevent movement of the larvae during imaging. The larvae were let to acclimatize for 30 min prior to imaging. FRAP experiments were performed at room temperaturte using Zeiss LSM 2MP multiphoton microscope with 20X water objective of 1.0 numerical aperture and Chameleon Ti-Sapphire laser (Coherent). The acquisition region was 116 × 116 pixels (28.34 μm$^2$), and interval between scanning was 0.35 s at a pixel dwell of 1.76 μs. Four circular ROI of 10 × 10 pixels encircling the synapses were selected for each larva, 3 of them for bleach and one for control. Another ROI of similar size was chosen outside the synapses for background quantification. 20 prebleach images were taken and bleach was performed at 75% laser power for 15 iterations and 200 postbleach images were taken.

For live imaging of synaptic OXT-EGFP, embryos were mounted and imaged as described above except for time-lapse parameters. Laser was used at 940 nm 2.5%. The acquisition region was 128 × 64 pixels (14.17 μm x 7.08 μm) and interval between scanning was 1 s at a pixel dwell of 3.15 μs. Two circular ROI of 10 × 10 pixels encircling the synapses were selected for each larva, one for bleach and one for control. FRAP were performed at different Z-stacks covering neurohypophysis. Another ROI of similar size was chosen outside the synapses for background quantification. 20

prebleach images were taken and bleach was performed at 75% laser power and 300 postbleach images were taken. Fluorescence images were drift corrected in Image/Fiji using TurboReg plugin (*Thévenaz et al., 1998*).

The FRAP values were analysed using EasyFRAP software with full scale normalization to account of difference in synaptic OXT-EGFP fluorescence (*Rapsomaniki et al., 2012*). The extracted data were analysed using custom written R-codes (See Statistical Analysis).

## Visualization of synaptic F-actin

For dual color three-dimensional STORM imaging, 5-dpf Tg(*oxt*:EGFP) or Tg(*oxt*:OXTSP-EGFP-OXT) embryos were immunostained with anti-OXT (1:200) or anti-EGFP (1:200) as described above. For visualization of actin and OXT, Tg(*oxt*:gal4; UAS:Lifeact-EGFP) larvae were fixed and stained with primary antibodies anti-Neurophysin (PS45) and anti-EGFP at 1:200. Secondary antibody (Alexa 647 or 568, Invitrogen) was used at a concentration at 1:2000. Samples were mounted ventrally without the jaws, and soaked in imaging buffer (50 mM 2-mercaptoethanol, 50 mM Tris-HCl (pH 8.0), 10 mM NaCl, 10% (w/v) glucose), 1x Gloxy for 15 min prior to imaging. 50X Gloxy buffer was made by making 8440 AU glucose oxidase (Sigma #G2133, 50 KU) and 70200 AU catalase (Sigma C40, 100 mg) in 50 mM Tris (pH 8.0) and 10 mM NaCl. Samples were imaged using Vutara SR-200 super-resolution microscope (Bruker). AlexaFluor647 was excited with 640 nm laser (power range of 4–9 kW/cm$^2$), AlexaFluor568 was excited with 561 nm laser (6 kW/cm$^2$) and 405 nm activation laser power was ramped slowly to maintain optimal single-molecule density. Images were recorded using a 60x, NA 1.2 water immersion objective (Olympus) and Evolve 512 EMCCD camera (Photometrics) with gain set at 50, frame rate at 50 Hz. Total number of frames acquired was 6000 per labeling dye. Data were analyzed by the Vutara SRX software.

To visualize F-actin using an alternate approach, Tg(*oxt*:gal4; *oxt*:OXTSP-EGFP-OXT) embryos were micro-injected with 10xUAS:mCherry-Utrophin-CH at similar conditions mentioned above. 8-dpf larvae were fixed in 4% PFA and imaged using confocal microscopy as mentioned above.

## TEM imaging

For TEM imaging of the neurohypophyseal axons and synapses, transgenic Tg(*oxt*:EGFP) zebrafish larvae (five dpf) were first anesthetized with tricaine and small incision was made on the dorsal part of the scalp, followed by immediate fixation with fixative buffer (4% PFA, 0.2% glutaraldehyde, 0.1 M cacodylate and 5 mM CaCl$_2$) overnight. The fixed larvae were then embedded in 3.4% Noble Agar (DIFCO) and sectioned using vibratome (OTS-4000, Electron Microscopy Sciences, Hatfield, PA). EGFP-positive slices (~200 μm thick), containing the neurohypophysis were washed in 0.1 M cacodylate buffer and incubated in 1% osmium tetraoxide, 0.5% potassium dichromate, 0.5% potassium hexacyanoferrate in 0.1 M cacodylate for 1 hr. Samples were rinsed in 0.1 M cacodylate and then in double distilled H$_2$O. Next, the samples were incubated in 2% uranyl acetate for 1 hr and covered with aluminum foil. Afterwards, samples were dehydrated in ethanol series and infiltrated for 5–7 days at RT in increasing concentration of Epon with ethanol. The Epon-infiltrated samples, were polymerized in at 60°C for 48 hr. Ultrathin sections (60–80 nm) mounted on a mesh grids (Electron Microscopy Sciences, USA) supported with carbon-coated nitrocellulose film. The ultrathin sections double stained with 2% uranyl acetate in ddH2O and Reynolds lead citrate (*Reynolds, 1963*) and imaged by Tecnai T12 electron microscope operating at 120 kV, utilizing an ES500W Erlangshen CCD camera (Gatan, UK) or an Eagle 2K × 2K CCD camera (FEI).

## Western blot

To validate the expression of OXT-EGFP fusion, adult pituitary of transgenic reporter Tg(*oxt*:SP-EGFP-OXT) were dissected and protein extracts were isolated. As controls, Tg(*oxt*:EGFP) pituitary extracts were isolated. Western blotting was performed on isolated protein extracts using anti-EGFP antibodies at 1:1000.

## Statistical analysis

Custom-written R codes were used for analysis of axonal swelling and OXT puncta data. For volume analysis, the values were log-transformed and the mean values were calculated for individual fish. The resulting mean values were back-transformed and statistical test were performed between

different genotypes or perturbations. To test for gaussian distribution of the data, Shapiro-Wilk normality test were performed. For comparison of 2 groups, Student's t-test or Mann-Whitney test were performed. For more than 2 groups, ANOVA or Kruskal-Wallis test were performed. Effect size was calculated using Cohen's d test.

For statistical analyses of FRAP experiments, in order to account the for inter-synapse variability within each animal as well as variability arising from differences between animals both within and between different genotypes (+/+ vs -/-), we generated a linear mixed effects model with time (repeated measure) and genotype as fixed effects, and synapse and fish ID as random effects. Each synapse and each fish were assigned random intercepts.

The correlation between the mean log-transformed ($\log_{10}$) fluorescence value of OXT puncta and EGFP axonal swelling was tested with a linear regression model (ANCOVA), accounting for the effect of EGFP axonal swelling, and individual fish.

## Acknowledgements

We thank Roy Hofi for animal care; Einav Wircer and Preethi Rajamannar for assisting in transgenic experiments; Harold Gainer (NINDS, Bethesda, USA) for the PS45 antibodies; Joshua Bonkowsky (Univ. of Utah, USA) for the Slit3-emEGFP construct and *robo2* mutant fish; Noaki Mochizuki (National Cerebral and Cardiovascular Center, Osaka, Japan) for Cdc42-related reagents; Wiebke Herzog (Univ. of Muenster, Germany) for the transgenic UAS:Lifeact-EGFP fish, Claire Wyart (ICM Institute, Paris, France) for the transgenic UAS:BoTxLCB-EGFP fish; Mary Hallaron (Univ. of Wisconsin, USA) for the mCherry-Utrophin-CH plasmid; Lior Appelbaum (Bar-Ilan Univ., Israel) for the UAS: Syp-EGFP fish; Nitzan Konstantin for English editing and Mike Ludwig (Univ. of Edinburgh) and Masha Prager-Khoutorsky (McGill Univ., Canada) for comments and discussions. SA was supported by Israel PBC-VATAT Fellowship and by Koshland Foundation. GL is supported by the Adelis Metabolic Research Fund and is an incumbent of the Elias Sourasky Professorial Chair.

## Additional information

### Funding

| Funder | Grant reference number | Author |
|---|---|---|
| Israel Science Foundation | 1511/16 | Savani Anbalagan<br>Janna Blechman<br>Michael Gliksberg<br>Ludmila Gordon<br>Gil Levkowitz |
| Israel Science Foundation | 2137/16 | Savani Anbalagan<br>Janna Blechman<br>Michael Gliksberg<br>Ludmila Gordon<br>Gil Levkowitz |
| Minerva Foundation | Minerva Stiftung | Savani Anbalagan<br>Janna Blechman<br>Michael Gliksberg<br>Ludmila Gordon<br>Gil Levkowitz |
| United States-Israel Binational Science Foundation | 2017325 | Michael Gliksberg<br>Gil Levkowitz |

The funders had no role in study design, data collection and interpretation, or the decision to submit the work for publication.

### Author contributions

Savani Anbalagan, Conceptualization, Data curation, Formal analysis, Validation, Visualization, Methodology, Writing—original draft, Writing—review and editing; Janna Blechman, Data curation, Formal analysis, Methodology, Writing—review and editing, Generated transgenic constructs, transgenic lines and performed in situ hybridization and biochemical experiments; Michael

Gliksberg, Formal analysis, Methodology, Writing—original draft, Writing—review and editing, Performed quantification of neurohypophyseal synapses and OXT puncta; Ludmila Gordon, Investigation, Visualization, Methodology, Performed TEM imaging; Ron Rotkopf, Formal analysis, Methodology, Writing—review and editing, Assisted in 'R' statistical analysis; Tali Dadosh, Visualization, Methodology, Writing—review and editing, Performed STORM imaging; Eyal Shimoni, Visualization, Methodology, Performed TEM imaging; Gil Levkowitz, Conceptualization, Resources, Supervision, Funding acquisition, Writing—original draft, Project administration, Writing—review and editing

### Author ORCIDs
Gil Levkowitz  https://orcid.org/0000-0002-3896-1881

### Ethics
Animal experimentation: Experiments involving zebrafish were approved by the Weizmann Institute's Institutional Animal Care and Use Committee (protocol #27220516).

### Decision letter and Author response
Decision letter https://doi.org/10.7554/eLife.45650.sa1
Author response https://doi.org/10.7554/eLife.45650.sa2

---

## Additional files

### Supplementary files
• Source code 1. R script for data extraction: R script to extract animal-wise neurohypophyseal synaptic parameters from the Volocity software-derived batch output files. The script generates tables with parameters such as animal ID, synapse or puncta ID, volume etc. In this manuscript, the script has been used for all the analysis in which number, synaptic OXT content and fluorescence values has been analyzed (*Figures 1–5* and *7*).

• Source code 2. R script for data transformation: R script to log-transform the neurohypophyseal synaptic volume data. In this manuscript, the script has been used for all the analysis in which number, synaptic OXT content and fluorescence values has been analyzed (*Figures 1–5* and *7*).

• Source code 3. R script for data analysis: R script to analyze log-transformed neurohypophyseal synaptic data. The mean number and volume of neurohypophyseal synapse and OXT puncta will be generated upon execution of the script. In this manuscript, the script has been used for all the analysis in which number, synaptic OXT content and fluorescence values has been analyzed (*Figures 1–5* and *7*).

• Source code 4. R script for Lifeact-EGFP FRAP data analysis: R script to analyze neurohypophyseal synapse. Lifeact-EGFP FRAP data by linear-mixed model (*Figure 4D*).

• Source code 5. R script for OXT-EGFP FRAP data analysis: R script to analyze neurohypophyseal synapse. OXT-EGFP FRAP data by linear-mixed model (*Figure 6I*).

• Source code 6. R script for correlation analysis: R script to analyze and plot neurohypophyseal synaptic. EGFP vs OXT fluorescence by linear regression (*Figure 7*).

• Transparent reporting form

### Data availability
All source data files and codes were uploaded. Source data as summary tables is provided for all graphs and plots shown (Figures 1E; 2B-E and G-J, 3E-H and J,K,M,N, 4D-F, 5O,R, 6I, 7C,D). R codes used for data import and analysis is provided.

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
