## [Decision Letter]

[Editors’ note: this article was originally rejected after discussions between the reviewers, but the authors were invited to resubmit after an appeal against the decision.]

Thank you for submitting your work entitled "Robo2 regulates synaptic oxytocin content by affecting actin state" for consideration by *eLife*. Your article has been reviewed by three peer reviewers, and the evaluation has been overseen by a Reviewing Editor and a Senior Editor. The following individuals involved in review of your submission have agreed to reveal their identity: David Murphy (Reviewer #2).

Our decision has been reached after consultation between the reviewers. Based on these discussions and the individual reviews below, we regret to inform you that your work will not be considered further for publication in *eLife*.

As you can see from the comments enclosed below, all reviewers find your work of interest. However, two of the three reviewers raised substantial issues that, in their opinion, need to be addressed before publication can be considered. After consultation among all reviewers it was agreed that the amount of work required for addressing even only the most important issues exceeds that what can be done in the two months’ time period allowed for revision. For these reasons, the reviewers felt that the manuscript cannot be accepted at present.

*Reviewer #1:*

In the present study, the authors aim at deciphering mechanisms relevant for the function of oxytocinergic synapses. Specifically, they tested whether the Robo2-Cdc42-actin pathway controls oxytocin homeostasis at zebrafish synapses. The authors exploited various transgenic lines and combined live imaging and super-solution microscopy with sophisticated image analyses. They showed a defined spatial distribution of the actin probe Lifeact in oxytocinergic synapses and that manipulation of actin affects the organization of oxytocinergic synapses. Further, they concluded from their experiments that the Robo2-Cdc42 pathway controls F-actin turnover and vesicle mobility in oxytocinergic synapses. Based on my concerns phrased below, some of these conclusions are not valid. Moreover, the rather mild effects reported in several experiments question whether the Robo2-Cdc42 pathway is biologically relevant for oxytocin function.

1) Throughout the manuscript, most reference literature concerning the presynaptic functions of actin is rather old; definitely, authors have to i) include more recent literature, which reported important novel insights into actin's presynaptic functions, and ii) revise their manuscript (Introduction, Discussion) according to the findings reported in these studies.

2) To prove that BoTxLCB indeed affect oxytocinergic synapses, the hypertonic osmotic challenge paradigm should be performed in the presence of BoTxLCB.

3) The authors exploited Lifeact in the assumption to specifically probe all F-actin, thereby neglecting several drawbacks of this tool. First, Lifeact is known for its high affinity for monomeric actin. Second, Lifeact's binding to F-actin is dependent on the presence of specific actin regulators (e.g. cofilin). Third, high Lifeact levels can affect endogenous F-actin structures. Authors have to include additional approaches to faithfully study the spatial distribution of F-actin in oxytocinergic synapses.

4) It is not entirely clear to me, which of the data provided led authors to suggest that actin-coated vesicles were associated with the plasma membrane of oxytocinergic synapses? Authors have to include membrane-tagged reporters or a fluorescent volume marker in their experiments.

5) Based on the aforementioned drawbacks of Lifeact, this probe is not suitable for studying F-actin turnover by FRAP. Authors should instead use GFP-actin, which has been proven to be highly useful to image actin functions in living cells and organisms.

6) Instead of using a global approach to link actin to the function of oxytocinergic synapses (treatment of larvae with cytochalasin D), authors should consider a genetic approach allowing manipulation of the actin cytoskeleton specifically in oxytocinergic neurons. Cytochalasin D effects on oxytocinergic synapses reported in this study may be rather indirect.

7) The effects of Robo2 inactivation on Lifeact turnover and oxytocin vesicle mobility a rather low. Although authors found various parameters to be significantly different between controls and Robo2-deficient animals, it is unclear whether Robo2-dependent mechanisms are indeed relevant for oxytocin function and biology. Authors should include additional analyses to tackle this.

*Reviewer #2:*

Anbalagan and colleagues have use state-of-the-art transgenic and imaging techniques to explore the molecular mechanisms whereby neuropeptides accumulate at synaptic release sites in order to provide the necessary levels required to maintain homeostasis. To do this, they have exploited two tractable models. Firstly, they have focussed on the elaboration of the evolutionarily conserved neuropeptide hormone oxytocin (OXT), which is synthesised as part of the prepropeptide precursor in discrete neurosecretory cells, then processed and transported to release sites in the anatomically distinct neurohypophysis. Secondly, they exploited the optically transparent zebrafish larva to analyse the function of the OXT system at single synapse resolution. The experiments are conceptually well designed, and have been carefully executed. The data produced is robust and aesthetically pleasing, and has been interpreted appropriately. The conclusions reached are important and completely new; the authors have discovered a novel signalling pathway that links Roundabout-2 (Robo2) receptor signalling to Cdc42 inactivation which, in turn, increases synaptic OXT content.

My only concerns relate to the narrative of the manuscript, which need some attention to improve the presentation of this otherwise excellent story.

The Introduction should be rewritten to provide a more cogent rational for the subsequent studies. The first sentence of the first paragraph should be followed by paragraph 3. This should be followed by an enunciation of a hypothesis that incorporates a possible role for the axonal cytoskeleton and Robo signalling in synaptic OXT content.

The work raises some interesting and important questions. This is a comprehensive and self-contained piece of work, and I am not suggesting that these issues need to be experimentally addressed. That said, the Discussion should be modified to take these points into consideration.

Firstly, the authors should speculate about the identity and source of the cognate ligand for the Robo2 receptor. Is there any evidence that secreted or membrane-bound Slit-family proteins are involved?

Secondly, the authors should integrate their findings into a more general discussion of excitation-synthesis-secretion coupling in OXT producing cells.

The Discussion ends rather abruptly. There should be a final paragraph that summarises the main findings.

*Reviewer #3:*

This manuscript presents experiments in zebrafish to test the relationships between of actin, the Robo pathway and the oxytocin (OXT) content of neurohypophyseal axons. The experiments show that manipulations of actin, Robo deficiency and Cdc42 dominant active alter these axons and have mild effects on their OXT content.

Major issues:

1) This study does not use the best possible approaches to support the main conclusions. The initial screen for changes in OXT content in axons uses light microscopy (dSTORM), which is good to first identify possible changes in OXT content, but this is not followed up by in depth analyses of individual synapses by electron microscopy (EM). Therefore, the main message remains vague and some aspects are not supported by strong enough data: The claim (in the title) that OXT accumulates 'in synapses' needs EM. Throughout the manuscript, the authors use an axonal filler to define "synapses". Hence, altered OXT levels can be anywhere in the axons (e.g. vesicles being transported through the axon). And changes in axon organization may explain the main findings (see below). Claims about vesicles "at the site of release" (Abstract) are not supported by data.

The conclusion that "the OXT content" is altered is vague. Are there fewer vesicles per terminal, fewer terminal axon branches or filopodia, fewer varicosities per axon length, smaller vesicles, less OXT per vesicle etc.? The term 'actin state' (Title) is also vague. What does that mean? Are release sites or varicosities larger?

One of the authors is from the "Electron Microscopy Unit" of the institute. Why is there no good EM to substantiate and better define the main conclusions?

2) The effects of Robo deficiency are very mild: a 10% on LifeAct dynamic fraction and similar (opposite) effect on stable fraction. Changes in OXT "accumulation" are also small. These effects may be indirect and non-specific. Might they even be due to different genetic backgrounds between control and Robo-/- or Cdc42 fish? Certainly, claims like "Robo2 is required…." are misplaced.

3) The effects of Cyto-D and Robo deficiency might be indirect and non-specific. In Figure 2 it is clear that changes in OXT content follow changes in total GFP signal (axonal filler). The effects of Cyto-D and Robo deficiency might primarily affect axonal organization (fewer terminal branches, filopodia, different axon diameter, different microtubule organization etc.). The mechanism of OXT accumulation in synapses may not be affected by Cyto-D, Robo deficiency or Cdc42.

---

## [Author Response]

[Editors’ note: the author responses to the first round of peer review follow.]

Thank you and the reviewers for your constructive feedback on our manuscript entitled, "Robo2 regulates synaptic oxytocin content by affecting actin state". We have carefully read and discussed all of the reviewers' comments, and in particular the issues which need to be addressed before publication. In your decision letter you wrote that the reviewers feel that addressing the most important issues will exceed what can be done in the two months’ time period allowed for revision. However, since the initial submission of the manuscript in January, we have gathered and analysed new data which directly address the concerns raised by the reviewers and therefore we trust that we can provide a revised manuscript within the allotted time.

Reviewer #1:[…]1) Throughout the manuscript, most reference literature concerning the presynaptic functions of actin is rather old; definitely, authors have to i) include more recent literature, which reported important novel insights into actin's presynaptic functions, and ii) revise their manuscript (Introduction, Discussion) according to the findings reported in these studies.

We have revised the Introduction (paragraph two) and Discussion (paragraph two) to include recent works on actin’s presynaptic functions.

2) To prove that BoTxLCB indeed affect oxytocinergic synapses, the hypertonic osmotic challenge paradigm should be performed in the presence of BoTxLCB.

We have performed the requested experiment and now show that the increased synaptic OXT level we observed in BoTxLCB transgenic larvae under naïve conditions, is maintained following hypertonic challenge (new Figure 2J). This result is consistent with inhibition of OXT synaptic release by BoTxLCB.

3) The authors exploited Lifeact in the assumption to specifically probe all F-actin, thereby neglecting several drawbacks of this tool. First, Lifeact is known for its high affinity for monomeric actin. Second, Lifeact's binding to F-actin is dependent on the presence of specific actin regulators (e.g. cofilin). Third, high Lifeact levels can affect endogenous F-actin structures. Authors have to include additional approaches to faithfully study the spatial distribution of F-actin in oxytocinergic synapses.

We agree with the reviewer that Lifeact has several drawbacks to probe F-actin. We have toned down our statement regarding the use of Lifeact to visualize spatial distribution of F-actin in oxytocinergic synapses (subsection “Disruption of F-actin affects synaptic OXT neuropeptide content” and Discussion paragraph five). As the reviewer suggested we now include an additional approach to detect F-actin in oxytocinergic synapses by using the Calponin domain of Utrophin fused to mCherry (UAS:mCherry-Utrophin-CH), which is a more-specific F-actin probe. We expressed mCherry-Utrophin-CH in OXT neurons in combination with our Tg(*oxt*:oxt-SP-EGFP) that labels OXT vesicles in synapses and found that mCherry-Utrophin-CH co-localizes with OXT-loaded synaptic vesicles (new Figure 6F). This result further supports our original suggestion that F-actin is enriched near OXT synaptic vesicles.

4) It is not entirely clear to me, which of the data provided led authors to suggest that actin-coated vesicles were associated with the plasma membrane of oxytocinergic synapses? Authors have to include membrane-tagged reporters or a fluorescent volume marker in their experiments.

The reviewer is correct. We have removed this statement.

5) Based on the aforementioned drawbacks of Lifeact, this probe is not suitable for studying F-actin turnover by FRAP. Authors should instead use GFP-actin, which has been proven to be highly useful to image actin functions in living cells and organisms.

As the reviewer suggested, we expressed GFP-actin specifically in OXT neurons, however this led to hypophyseal axon guidance defects in our in vivoexperimental system and precluded further analysis; these morphological abnormalities were not observed following OXT-specific expression of Lifeact-GFP. We also attempted to utilize the mCherry-Utrophin-CH probe, however, we were unable to perform FRAP experiments with this reagent due to the deep location, long wave length and high laser power required to perform mCherry FRAP with two-photon.

We agree with the reviewer about the drawbacks of Lifeact to study F-actin turnover by FRAP. We discuss Lifeact shortcomings in the revised manuscript (subsection “Disruption of F-actin affects synaptic OXT neuropeptide content” and Discussion paragraph five). Having said that, we still maintain that this experiment is informative in order to show that synaptic actin dynamics is altered in *robo2* mutants. Furthermore, we have now provided new information to demonstrate the causal effects of changes in actin dynamics on synaptic OXT content (see our reply to comment number 6 of this reviewer).

6) Instead of using a global approach to link actin to the function of oxytocinergic synapses (treatment of larvae with cytochalasin D), authors should consider a genetic approach allowing manipulation of the actin cytoskeleton specifically in oxytocinergic neurons. Cytochalasin D effects on oxytocinergic synapses reported in this study may be rather indirect.

As suggested by this reviewer, to specifically test the role of actin in oxytocinergic synapses, we undertook a genetic approach by over-expressing Cofilin-EGFP specifically in oxytocin neurons using our transgenic tools (*oxt*:Gal4; UAS:Cofilin-EGFP). Cofilin is a member of ADF family of actin-binding proteins that promote F-actin depolymerization. We observed that expression of EGFP did not affect number and size of OXT puncta (new Figure 3J-K), however, expression of Cofilin-EGFP led to decreased OXT content as measured by puncta volume (new Figure 3N). These results suggest that actin have a direct role in regulating synaptic OXT content.

7) The effects of Robo2 inactivation on Lifeact turnover and oxytocin vesicle mobility a rather low. Although authors found various parameters to be significantly different between controls and Robo2-deficient animals, it is unclear whether Robo2-dependent mechanisms are indeed relevant for oxytocin function and biology. Authors should include additional analyses to tackle this.

In response to this comment and a similar remark made by reviewer #3, we calculated effect sizes (Cohen's D; Cohen, 2013) for the differences in stable vs. dynamic Lifeact fractions, and now show that the differences are in fact, quite large. Thus, when analyzing Lifeact stable vs. dynamic fractions in *robo2*+/+ compared to *robo2*-/-, Cohen's D is equal to 0.67 (subsection “Robo2 regulates synaptic actin dynamics”).

We would like to emphasize that the significant oxytocin vesicle mobility defect (Genotype x Time is p<0.0001) in *robo2* -/- mutant is for a period of 5 minutes of live imaging the synapses. The cumulative effect of this over a longer period can lead to a larger effect. This is exemplified in reduced OXT content in *robo2*-/- for which we also calculated Cohen’s D of 0.82 indicating large effect size (new Figure 5D) (subsection “Slit3-Robo2 signalling regulates synaptic OXT content”).

Finally, we respect the reviewer comment regarding analysis of oxytocin function and biology in *robo2* mutants, however we feel that measuring oxytocin-relevant physiological and behavioural outcomes are beyond the scope of this manuscript.

Reviewer #2:[…]My only concerns relate to the narrative of the manuscript, which need some attention to improve the presentation of this otherwise excellent story.The Introduction should be rewritten to provide a more cogent rational for the subsequent studies. The first sentence of the first paragraph should be followed by paragraph 3. This should be followed by an enunciation of a hypothesis that incorporates a possible role for the axonal cytoskeleton and Robo signalling in synaptic OXT content.

We have now extensively revised the Introduction and also incorporated the suggestions by the reviewer.

The work raises some interesting and important questions. This is a comprehensive and self-contained piece of work, and I am not suggesting that these issues need to be experimentally addressed. That said, the Discussion should be modified to take these points into consideration.Firstly, the authors should speculate about the identity and source of the cognate ligand for the Robo2 receptor. Is there any evidence that secreted or membrane-bound Slit-family proteins are involved?

The role of cognate ligand for the Robo2 receptor is now demonstrated in the new Figure 5F-R. In short, we recently identified that cognate ligand of Robo2, *slit3* is highly expressed in the hypophyseal tissue (Anbalagan et al., 2018). We have performed additional genetic gain/loss-of-function experiments to study the role of Slit3 in regulating synaptic OXT content. Knockdown of *slit3* led to reduced OXT content similar to the *robo2* mutant phenotype (new Figure 5I). Conversely, ectopic expression of Slit3 in the neighbouring POMC-positive cells led to increased OXT content, but only in those neurohypophyseal synapses, which were in very close proximity to Slit3-overexpressing cells (new Figure 5L-R).

These results suggest that Slit3-Robo2 signalling regulate synaptic OXT levels.

Secondly, the authors should integrate their findings into a more general discussion of excitation-synthesis-secretion coupling in OXT producing cells.

We have added a discussion paragraph regarding the possible role of Slit3-Robo2 signalling and excitation-secretion coupling (Discussion paragraph seven).

The Discussion ends rather abruptly. There should be a final paragraph that summarises the main findings.

We added a concluding sentence to the Discussion.

Reviewer #3:[…]Major issues:1) This study does not use the best possible approaches to support the main conclusions. The initial screen for changes in OXT content in axons uses light microscopy (dSTORM), which is good to first identify possible changes in OXT content, but this is not followed up by in depth analyses of individual synapses by electron microscopy (EM). Therefore, the main message remains vague and some aspects are not supported by strong enough data: The claim (in the title) that OXT accumulates 'in synapses' needs EM. Throughout the manuscript, the authors use an axonal filler to define "synapses".

It is well accepted that the axonal varicosities/swellings of hypothalamic magnocellular OXT neurons, which converge into the neurohypophysis are bona fidesynaptic release sites that store OXT-containing LDCV and release them upon physiological demand (Miyata et al., 2001; Wittkowski and Brinkmann, 1974). We agree with this reviewer that this has not been demonstrated in zebrafish larvae. To address the reviewer concerns (detailed below) we now provide new data from independent experimental approaches:

a) “The claim (in the title) that OXT accumulates 'in synapses' needs EM.” As the reviewer requested we teamed up with Dr. Eyal Shimoni from our institute’s electron microscopy (EM) unit and performed transmitted electron microscopy (TEM) in which we used our transgenic *oxt*:EGFP reporter as a fluorescent landmark to localize neurohypophyseal axonal swellings. We now show that electron-dense large dense core vesicles containing neuropeptides are indeed enriched in larval zebrafish neurohypophyseal synapses (Figure 1A-C). These synapses also have classical neurosecretory characteristics described in other species, including membrane fusion event during the exocytosis of LDCVs content (Figure 1B’) into the perivascular space, as well as vesicle recycling events (Figure 1C’). Such exocytotic activity have been previously reported in neurohypophyseal synapses (Boudier, 1974; Buma and Nieuwenhuys, 1987; Crosnier et al., 2010; Damer and Creutz, 1994; De Camilli, 1995; Douglas, 1970; Douglas, 1973; Hayashi et al., 1994).

b) “The authors use an axonal filler to define “synapses”.” We provide additional evidence for the synaptic identity of OXT-positive neurohypophyseal axonal swellings using the transgenic synaptic vesicle reporter, Synaptophysin-EGFP. We now show that practically all analysed oxytocin-positive puncta co-localize with Synaptophysin (mean weighted correlation coefficient = 0.87; n=5 larvae new Figure 1D,E).

Notably, larval zebrafish neurohypophysis is very small (~20x50μm) and is hard to localize for EM analysis, hence it is not practical to quantify neuropeptide content using EM. Having demonstrated the synaptic identity of OXT-positive swellings we took advantage of the transparent zebrafish larvae combined with precise genetic tools and reporters to quantify OXT synapses using confocal images on immunostained fixed samples and multi-photon based imaging on live transgenic larvae.

Hence, altered OXT levels can be anywhere in the axons (e.g. vesicles being transported through the axon). And changes in axon organization may explain the main findings (see below). Claims about vesicles "at the site of release" (Abstract) are not supported by data.

We agree that changes in axon organization may explain the main findings. To address this point, we quantified the number and size of the neurohypophyseal synapses as well as the numbers of projecting axons. We observe no differences in those parameters following genetic perturbation of *slit3* and *robo2* as well as following OXT-specific gain-of-function of the actin regulator cofilin (new Figure 3 and 5).

In view of the new data presented above we maintain that we are analysing OXT levels and synaptic vesicles dynamics at the site of release.

The conclusion that "the OXT content" is altered is vague. Are there fewer vesicles per terminal, fewer terminal axon branches or filopodia, fewer varicosities per axon length, smaller vesicles, less OXT per vesicle etc.?

As shown in our EM image (Figure 1B,C), neurohypophyseal synapses contain multiple large dense core vesicles (LDCVs).

Analysis of OXT content is based on immune-reactive OXT puncta that colocalized with genetically EGFP-tagged synapses. Due to the elaborate 3D structure of neurohypophyseal projections we are unable to reliably quantify axon branches or filopodia/varicosities per axon. Our real-time analysis of synaptic OXT vesicles accumulation suggests that the decreased volume of OXT puncta in neurohypophyseal synapses, following various perturbations of actin dynamics is most likely due to reduced OXT content in the synapses (i.e. fewer OXT containing LDCVs).

The term 'actin state' (Title) is also vague. What does that mean? Are release sites or varicosities larger?

We have modified the term ‘actin state’ to ‘actin dynamics’.

We did not observe changes in varicosities sizes following *robo2-/-, slit3* knockdown (new Figure 5A-J) as well as following OXT-specific gain-of-function of cofilin (new Figure 3-N). Thus, we surmise that the outcome of these actin perturbations is not reflected in parameters relating to the shape of the cell (changes in varicosity size, branch points, axon guidance, etc.), but rather to changes in the dynamics of the actin cytoskeleton which affects OXT-vesicle mobility/accumulation in neurohypophyseal synapses.

One of the authors is from the "Electron Microscopy Unit" of the institute. Why is there no good EM to substantiate and better define the main conclusions?

As mentioned above, we teamed up with an EM expert, Dr. Eyal Shimoni, and now provide new ultrastructure images to substantiate our claims. Dr. Shimoni was added as a co-author to the revised manuscript.

2) The effects of Robo deficiency are very mild: a 10% on LifeAct dynamic fraction and similar (opposite) effect on stable fraction. Changes in OXT "accumulation" are also small. These effects may be indirect and non-specific. Might they even be due to different genetic backgrounds between control and Robo-/- or Cdc42 fish? Certainly, claims like "Robo2 is required…." are misplaced.

Control for all experiments involving the effects of Robo2 were performed using larvae derived from in-cross of *robo2*+/- fish (i.e. full siblings), ruling out that changes in Lifeact dynamic and OXT accumulation are due to differences in genetic background.

Furthermore, as mentioned in our response to comment #7 of reviewer #1, we now provide effect sizes (Cohen's D) for all relevant analyses and show that the effect sizes of actin perturbations on synaptic OXT content are in fact, quite large. Thus, in the case of the Lifeact stable vs. dynamic fractions (new Figure 4E,F), Cohen's D is equal to 0.67, and in the case of accumulation of OXT content in synapses (new Figure 5D) Cohen’s D is 0.82. For a discussion of statistical analyses of FRAP experiments, please see our reply to the specific comment by this reviewer regarding statistical treatment of these types of experiments (final comment).

Lastly, we agree with the reviewer that other Robo2-independent actin-regulatory mechanisms can regulate synaptic actin dynamics. We have therefore changed the wordings “Robo2 is required…” to “Robo2 regulates…” throughout the manuscript.

3) The effects of Cyto-D and Robo deficiency might be indirect and non-specific: In Figure 2 it is clear that changes in OXT content follow changes in total GFP signal (axonal filler). The effects of Cyto-D and Robo deficiency might primarily affect axonal organization (fewer terminal branches, filopodia, different axon diameter, different microtubule organization etc.). The mechanism of OXT accumulation in synapses may not be affected by Cyto-D, Robo deficiency or Cdc42.

We agree with the reviewer that the effect of Cyto-D can be also due to changes in the size of neurohypophyseal synapses. As mentioned in our reply to reviewer #1 (comment #6), we now provide additional experiments to directly test the role of F-actin and CDC42 in regulating synaptic OXT content. Thus, we undertook an alternative genetic approach to conditionally express either the Actin depolymerizing factor Cofilin or constitutively-active (i.e. GTP-bound) mutant form of CDC42, specifically in OXT neurons. Both of these cell type specific perturbations of Actin dynamics led to decreased OXT content (new Figure 3I-N and new Figure 7—figure supplement 1).